# Claudin-23 reshapes epithelial tight junction architecture to regulate barrier function

Arturo Raya-Sandino [1,5], Kristen M. Lozada-Soto [1,5], Nandhini Rajagopal[2], Vicky Garcia-Hernandez [1], Anny-Claude Luissint [1], Jennifer C. Brazil[1], Guiying Cui[3], Michael Koval [4], Charles A. Parkos [1], Shikha Nangia [2] ✉ & Asma Nusrat [1] ✉

Claudin family tight junction proteins form charge- and size-selective paracellular channels that regulate epithelial barrier function. In the gastrointestinal tract, barrier heterogeneity is attributed to differential claudin expression. Here, we show that claudin-23 (CLDN23) is enriched in luminal intestinal epithelial cells where it strengthens the epithelial barrier. Complementary approaches reveal that CLDN23 regulates paracellular ion and macromolecule permeability by associating with CLDN3 and CLDN4 and regulating their distribution in tight junctions. Computational modeling suggests that CLDN23 forms heteromeric and heterotypic complexes with CLDN3 and CLDN4 that have unique pore architecture and overall net charge. These computational simulation analyses further suggest that pore properties are interaction-dependent, since differently organized complexes with the same claudin stoichiometry form pores with unique architecture. Our findings provide insight into tight junction organization and propose a model whereby different claudins combine to form multiple distinct complexes that modify epithelial barrier function by altering tight junction structure.

Epithelial tissues create a physical barrier between the external environment and underlying mucosal tissue compartments while facilitating selective transport of solutes and ions. Epithelial barrier function is regulated by intercellular junctions encompassing the tight junction (TJ), subjacent adherens junction, and desmosomes. The TJ is the most apical junctional complex bringing adjacent plasma membranes into close proximity in order to seal the paracellular space between cells and regulate the passage of ions and molecules through the paracellular pathway[1–4]. As such, proper function of epithelial TJs is essential for establishment and maintenance of tissue homeostasis.

Claudins (CLDNs) represent a family of cell-cell adhesion proteins that polymerize to form the backbone of TJ strands. It has been well established that CLDNs regulate TJ selectivity through formation of single-pore paracellular channels that conduct small ions and

molecules in a charge- and size-specific manner[5,6]. Structurally, CLDN proteins have four transmembrane helices, a short intracellular $NH_2$ terminus along with a longer COOH terminus, and two extracellular segments (ECS). Residues in ECS1 are known to be crucial for the establishment of channel properties and mediating ion charge selectivity through the "pore pathway"[5,7–13]. Pore pathway selectivity is further defined by the expression levels of channel-forming CLDNs and barrier-forming CLDNs at the TJ. To date, CLDN2, −10b, −15, −16 and −21 have been shown to form cation-selective paracellular channels, whereas CLDN10a and −17 are known to form anion-selective paracellular channels[5]. Barrier-forming claudins, including CLDN1, −3, −5, and −11[12,14–16], act to restrict the paracellular flux of ions and small solutes. Other family members, such as CLDN4, −7, and −8 display context-dependent barrier regulation properties that appear closely

[1]Department of Pathology, University of Michigan Medical School, Ann Arbor, MI, USA. [2]Department of Biomedical and Chemical Engineering, Syracuse University, Syracuse, NY, USA. [3]Department of Pediatrics, Emory + Children's Center for Cystic Fibrosis and Airways Disease Research, Emory University School of Medicine, Atlanta, GA, USA. [4]Departments of Medicine and Cell Biology, Emory University School of Medicine, Atlanta, GA, USA. [5]These authors contributed equally: Arturo Raya-Sandino, Kristen M. Lozada-Soto. ✉e-mail: snangia@syr.edu; anusrat@umich.edu

tied to the tissue in which they are synthesized as well as the expression of additional CLDNs[17–24]. Taken together these studies highlight the existence of sophisticated mechanisms of paracellular barrier function regulation by members of the CLDN family[5]. CLDNs can also be grouped into classic and non-classical CLDNs based on degree of sequence similarity. Compared to non-classical CLDNs, classical CLDNs display higher levels of sequence homology and similar TJ assembly mechanisms[25–28].

At the TJ plasma membrane, CLDNs associate side by side (in *cis*) to create linear polymers known as CLDN-based strands. *Cis* interactions that occur between the same CLDN family member are termed homomeric, whereas heteromeric interactions refer to those that exist between different CLDN family members. The attachment of several CLDN-based strands facilitates formation of elaborate networks that can vary in complexity and organization across different tissues. Alongside the frequency and density of TJ strands, the specific combination of barrier-forming and channel-forming CLDNs has profound effects on selective flux of ions, water, and small solutes across the paracellular space in the intestine. TJ strands are dynamic structures that undergo frequent remodeling[29,30]. The rapid and continuous turnover of CLDNs alongside breaking and annealing of strands regulates the strand network structure creating discontinuities that form the basis of a second paracellular pathway, known as the nonrestrictive or "leak pathway"[30–33]. During the process of strand break and repair, larger molecules can permeate the TJ via this leak pathway, albeit at lower levels and with less selectivity compared to transit via the pore pathway[34,35].

CLDNs expressed on opposing plasma membranes of neighboring cells can also associate head-to-head (in *trans*)[7,26,36,37]. While homotypic *trans* interactions between the same CLDN subtype are most common, and have been computationally modeled in several studies[38–42], *trans* interactions between different family members (heterotypic) have been reported to occur under highly specific circumstances[43,44]. For example, while CLDN3 can interact with CLDN1, CLDN2, and CLDN5[43,45,46], CLDN4 does not associate in *trans* with either CLDN3 or CLDN2[43,47]. The specificity and heteromeric compatibility of CLDNs is similar to that of other multimeric channels such as epithelium sodium channels (ENaC), formed by heteromultimeric membrane glycoproteins, and connexin-based gap junction channels[48,49]. TJ strand formation and stability, along with *trans* CLDN-CLDN interactions, are critical for the formation of paracellular CLDN channels. Additionally, recent progress has highlighted the importance of charged pore-lining residues in determining the geometry and ion charge selectivity of CLDN channels[50]. Together, the molecular complexity of the CLDN channels implies important functional consequences. However, how different CLDNs combine to alter channel architecture and the potential impact of such interactions on TJ permeability to ions and macromolecules has not been studied to date.

Combinatorial expression of CLDN family members occurs in an organ- and developmental stage-specific manner in order to regulate spatial barrier properties and create epithelial tissue microenvironments with individual paracellular permeability characteristics. In the colon, the epithelium transitions from a proliferative to a differentiated state every 5–6 days while maintaining tight barrier properties at the luminal surface[51,52]. During this dynamic life cycle, junctions are remodeled with intestinal epithelial cells (IECs) undergoing dynamic switches in CLDN expression and localization. IECs in the crypt base have increased expression of channel-forming CLDN2 and CLDN15, whereas surface epithelial cell populations are enriched for the prototypic barrier-forming CLDN3, as well as CLDN4 and CLDN7[51,53–56]. Differential expression of CLDNs allows for spatial regulation of barrier properties along the crypt-luminal axis rendering the base of intestinal crypts as "leaky" compared to the tighter surface epithelium.

Previous studies from our group identified *Cldn23* mRNA expression in colonic epithelial surface cells at the luminal surface[53].

CLDN23 is a non-classical claudin that has low sequence homology with classical CLDNs[25] (Supplementary Fig. 1a). Unique features of CLDN23 include a long cytoplasmic tail (111 amino acids) and two uncharged residues at sites within the ECSs predicted to determine ion charge selectivity of the channel[5]. Although previous work has demonstrated a potential role for CLDN23 during neoplastic transformation in gastric, pancreatic, and colorectal cancer[57–61]; its role in regulating epithelial tissue homeostasis has not been explored. However, given the abundant expression of *Cldn23* mRNA in luminal IECs[53], it is likely that this atypical CLDN plays an important role in regulating mucosal barrier function in the gut. Herein, using complementary biochemical and computational approaches, we present evidence that CLDN23 plays a non-redundant role in functionally strengthening epithelial barrier properties. We further report that CLDN23 recruits prototypic barrier-forming claudins, CLDN3 and CLDN4, to TJs in a manner that restricts paracellular permeability to ions and macromolecules and strengthens barrier function. Furthermore, in silico modeling supports these experimental results showing that CLDN23 interacts with CLDN3 and CLDN4 to restrict and block formation of paracellular pores, consistent with the impact of CLDN23 on TJ ion permeability.

Taken together, our results support an emerging concept that interactions between different CLDN family members underlie an elaborate mechanism of barrier regulation that can alter CLDN channel architecture and paracellular permeability properties of epithelial TJs. Increased understanding of how specific CLDN-CLDN interactions regulate barrier function is important in the context of inflammatory disorders where dysregulated CLDN expression is associated with disease pathogenesis.

## Results
### CLDN23 is differentially expressed in IECs along the crypt–luminal axis

To determine the expression of *CLDN23* mRNA in the colon, RNAScope in situ hybridization was performed in colonic tissue obtained from healthy human donors and C57BL/6 wild-type (WT) mice (from Jackson Labs). Epithelial expression of *CLDN23* mRNA was detected in cells along the length of the crypt-luminal axis, with a 3–4-fold increase in expression observed in luminal or surface IECs for both human and murine colonic tissue. These findings highlight predominant expression of CLDN23 in differentiated IECs (Fig. 1a, b). The spatial expression of CLDN23 protein was further analyzed using a novel rabbit polyclonal antibody raised against two unique regions of the C-terminal tail of murine CLDN23 (NH2-PRPPPKSYTNPMDVLEGEEK-COOH and NH2-GGSSSRSTRPCQNSLPCDSD-COOH), that share 72% and 79% identity with human CLDN23, respectively. Specificity of the CLDN23 antibody was validated by immunoblotting for CLDN23 expression in transfected HeLa cells that are otherwise claudin-null. As can be seen in Supplementary Fig. 2, the CLDN23 antibody recognized a single band at the predicted molecular weight (MW) for CLDN23 (-32 kDa) in CLDN23 expressing HeLa cells while not recognizing any protein targets in the WT HeLa cells. Since IECs in the crypt base express CLDN2 while luminal IECs express CLDN3 and CLDN4, we also generated HeLa cells individually expressing these specific CLDNs (Supplementary Fig. 2). Importantly, the CLDN23 antibody did not cross-react with CLDN2, CLDN3, or CLDN4 (Supplementary Fig. 2). Furthermore, immunofluorescence labeling and confocal microscopy of human and murine colonic tissues revealed a gradient of CLDN23 protein expression in IECs localized in the mid-region and at the luminal surface of crypts that correlated with *CLDN23* mRNA expression (Fig. 1c. d). At the cellular level, CLDN23 protein was predominantly localized to the apical aspect of the lateral plasma membrane corresponding to the location of the bicellular TJs (Fig. 1c, arrows).

To analyze the expression of CLDN23 in differentiating IECs in the crypt-luminal axis, we employed an in vitro model mimicking IEC

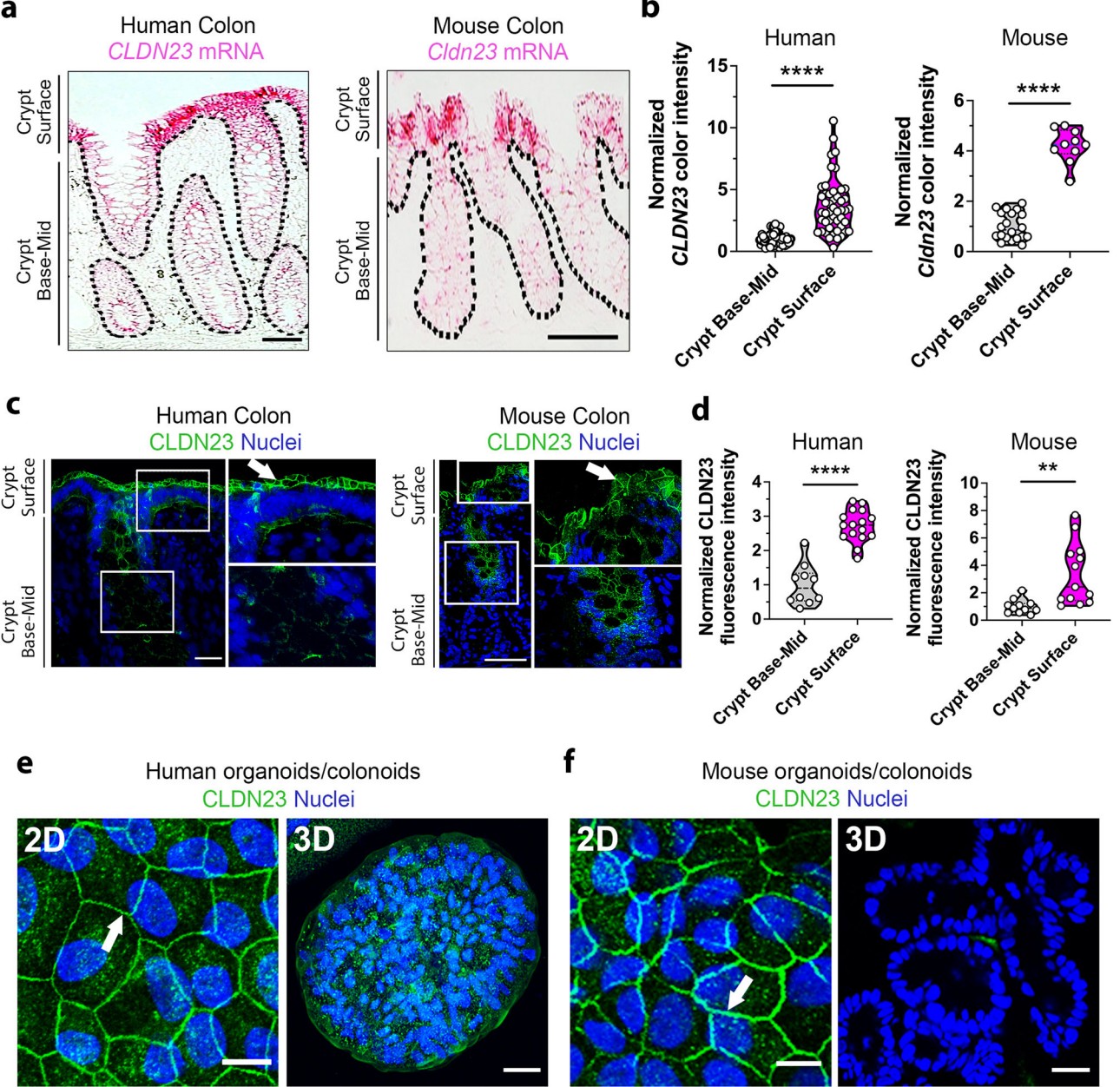

**Fig. 1 | CLDN23 is differentially expressed in IECs along the crypt-luminal axis.**
**a** *CLDN23* mRNA expression (pink) detected by RNAscope in situ hybridization in human (left) and C57BL/6 WT murine (right) colonic epithelial cells. Scale bar: 50 μm. **b** Histograms represent the color intensity of *CLDN23* mRNA staining at the base-mid and surface of individual colonic crypts in human (left) or murine (right) tissues. Each dot represents an individual crypt. Data are mean ± SD of 9 images from five biopsies from healthy human (64 crypts total) or WT murine colons (20 crypts total). Values were normalized to the intensity of the base-mid section of the crypt. ****$p \le 0.0001$; two-tailed Student's *t* test. **c** Confocal images showing CLDN23 protein (green) and nuclei (DAPI/blue) in healthy human (left) and C57BL/6 WT murine (right) colonic epithelium. Scale bar: 50 μm. **d** Histograms represent

normalized fluorescence intensity for CLDN23 at the base-mid and surface of individual colonic crypts in human (left) or mouse (right) tissues. Each dot represents an individual crypt. Data are mean ± SD of 9 images obtained from three healthy human colonic biopsies (16 crypts total) or three WT murine colonic tissues (13 crypts total). Values were normalized to the fluorescence of the base-mid section of the crypt. ****$p \le 0.0001$, **$p = 0.0025$; two-tailed Student's *t* test. Representative confocal images showing CLDN23 (green) and nuclei (DAPI/blue) in differentiated (2D) and undifferentiated (3D) colonoids derived from human (**e**) and C57BL/6 mouse (**f**) crypts. Scale bars: 10 μm (left) and 50 μm (right). CLDN23 expression in 3D colonoids with stem cell-like phenotypes (**e**, **f**, right), and at cell-cell contacts between differentiated cells in 2D monolayers (**e**, **f**, left/arrow).

differentiation. Primary culture IECs (colonoids) derived from healthy human colon and C57BL/6 WT mice were grown either as cysts in three-dimensional (3D) cultures that maintain a proliferative/stem-like state or as two-dimensional (2D) monolayers induced to differentiate for 1–3 days in growth factor-optimized medium[62]. As shown in Fig. 1e, f, CLDN23 protein expression was observed in cell-cell contacts between differentiated IECs in 2D colonoid monolayers derived from both healthy human and mouse colon (left panels). In contrast,

minimal CLDN23 expression was identified in undifferentiated 3D colonoids (Fig. 1e, f, right panels), suggesting an important role for CLDN23 in differentiated IECs. To expand on these observations, we next analyzed CLDN23 expression in an in vitro system of epithelial differentiation/junction maturation in model human intestinal epithelial cells (SKCO15 and T84) (Supplementary Fig. 3a)[63]. As was observed in 2D colonoids, increased expression of CLDN23 was observed upon differentiation of human intestinal epithelial cell lines.

In addition IEC differentiation was characterized by increased expression of the intestine-specific transcription factor CDX2[64], as well as upregulation of prototypic barrier-forming CLDN3 and CLDN4, and downregulation of channel-forming CLDN2 (Supplementary Fig. 3b, c). Furthermore, we observed that CLDN23 colocalizes with CLDN4 in differentiated murine colonoid monolayers (Supplementary Fig. 4c). In summary, these results demonstrate that CLDN23 is primarily expressed in differentiated colonic IECs.

## CLDN23 regulates intestinal epithelial barrier function

Since CLDN23 expression was identified at the most apical portion of the lateral plasma membrane, where the TJ resides, and given that surface IECs express several barrier-regulating CLDNs, we hypothesized that CLDN23 could function as a barrier-forming CLDN. To test this hypothesis, human model IEC cell lines with either augmented or silenced CLDN23 expression were generated. SKCO15 IECs were transduced with full-length human CLDN23 resulting in constitutive overexpression of CLDN23 (Fig. 2a). In parallel, endogenous CLDN23 expression in T84 IECs was silenced by viral transduction using two different sets of shRNA against CLDN23 (Fig. 2d). Of note, endogenously expressed CLDN23 protein in T84 IECs migrated with an apparent MW of 130 kDa, suggesting the formation of CLDN23 containing SDS-PAGE stable oligomers, as has been observed for other CLDNs[65,66]. Specificity of the 130 kDa protein band was confirmed by disappearance after shRNA mediated knockdown of CLDN23. Successful overexpression of CLDN23 in SKCO15 cells and knockdown in T84 cells was confirmed by immunoblotting (Fig. 2a, d). In keeping with immunoblotting results, immunofluorescence labeling analysis revealed uniform overexpression and knockdown of CLDN23 in model IEC lines (Supplementary Fig. 4a, b). In addition, immunofluorescence labeling analysis revealed colocalization of CLDN23 with TJ-localized CLDN4, demonstrating the expected CLDN23 distribution at the TJ (Supplementary Fig. 4a, b, arrows). Furthermore, cytoplasmic expression of CLDN23 was also observed in vesicle-like structures (Supplementary Fig. 4a, b, arrowheads).

To evaluate the contribution of CLDN23 to the regulation of barrier function in vitro, transepithelial electrical resistance (TEER) of human IEC monolayers with either enhanced or silenced CLDN23 expression was continuously measured on semipermeable inserts (transwells) by using an automated cell impedance monitoring system (Fig. 2b, e). SKCO15 cells over-expressing CLDN23 exhibited a sharp increase in TEER one day after plating with resistance values increasing to 150% of the rate of control cells by day 5 (Fig. 2b). In contrast, knockdown of CLDN23 expression in T84 IECs resulted in decreased resistance values at all time points measured at 5 days of confluency (45% for shRNA1 and 20% for shRNA2) (Fig. 2e). Of note, T84 IECs with >95% reduction in CLDN23 expression failed to increase their resistance values above 200 $\Omega \cdot cm^2$ (Fig. 2e), indicating an important role for CLDN23 in regulating IEC barrier formation. For further analysis of CLDN23-mediated barrier regulation, paracellular permeability to macromolecules (TD4 dextran and FD70 dextran) was assessed. CLDN23 overexpression in SKCO15 IECs resulted in a statistically significant 3-fold decrease in the passage of TD4 (Fig. 2c). In contrast, loss of CLDN23 expression resulted in significantly increased rates of paracellular permeability to TD4 (Fig. 2f). These data suggest that CLDN23 regulates IEC paracellular permeability through the non-restrictive pathway (also referred to as the 'leak pathway'). Of note, the flux rate of FD70 dextran was not affected by either increased or decreased expression of CLDN23 (Fig. 2c, f), indicating that monolayer integrity was intact and that paracellular transport to macromolecules was size-selective as expected.

To complement in vitro findings and evaluate the physiological role of CLDN23 in regulating IEC barrier function in vivo, we generated mice with tamoxifen-inducible IEC-specific deletion of CLDN23 ($Cldn23^{ERΔIEC}$) (Supplementary Fig. 5a). Specific knockdown of CLDN23

in IECs was confirmed by analysis of $Cldn23$ mRNA and CLDN23 protein expression (Fig. 2g). Immunofluorescence labeling of mouse colon confirmed complete depletion of IEC-expressed CLDN23 in tamoxifen-induced mice (Fig. 2h). Importantly, CLDN23 deletion did not result in any alterations or abnormalities to colonic mucosal architecture or any spontaneous inflammation (Supplementary Fig. 5b). The contribution of CLDN23 in regulating paracellular permeability to macromolecules in vivo was assessed using an exteriorized ileal loop model, whereby 4 kDa FITC-dextran (FD4) is allowed to passively permeate from the intestinal lumen into the serum[67–69] (Fig. 2i, left). FD4 fluorescence intensity in the serum was measured in control $Cldn23^{f/f}$ and $Cldn23^{ERΔIEC}$ mice and normalized to levels in control animals. Importantly, CLDN23 depleted mice had a statistically significant 2–3.5-fold increase in FD4 flux compared to control $Cldn23^{f/f}$ mice (Fig. 2i, right) highlighting a role for CLDN23 in regulating macromolecule flux across intestinal barriers.

To complement in vivo findings, murine colonoids from $Cldn23^{f/f}$ and $Cldn23^{ERΔIEC}$ mice were isolated and differentiated to induce CLDN23 expression. Immunofluorescence labeling of differentiated colonoid monolayers revealed near-total loss of CLDN23 expression in $Cldn23^{ERΔIEC}$-derived colonoids (Supplementary Fig. 4c). As was observed for T84 and SKCO15 IECs, $Cldn23^{f/f}$ colonoid monolayers exhibited linear CLDN23 staining along the plasma membrane consistent with TJ localization (Supplementary Fig. 4c, arrows). We also observed staining consistent with cytoplasmic distribution of CLDN23 in murine colonoids (Supplementary Fig. 4c, arrowheads). Analysis of TEER in colonoid-derived monolayers revealed a statistically significant 85% decrease in resistance in the absence of CLDN23 expression following 2 days of differentiation (Supplementary Fig. 6). Taken together these data highlight a non-redundant and important role for CLDN23 in regulating paracellular permeability to macromolecules and strengthening intestinal epithelial barrier function in vivo.

## CLDN23 influences CLDN3 and CLDN4 protein localization in the TJ plasma membrane

We next investigated whether CLDN23-mediated barrier function regulation is dependent on alterations in expression of other TJ-associated proteins. Immunoblotting of colonic mucosa from $Cldn23^{f/f}$ and $Cldn23^{ERΔIEC}$ mice did not reveal any differences in expression of CLDN2, CLDN3, or CLDN4 in $Cldn23^{ERΔIEC}$ compared to control mice (Fig. 3a). In addition, loss of CLDN23 has no obvious effect on expression of ZO1, a cytoplasmic TJ scaffolding protein known to bind to CLDNs (Fig. 3a). These results were further confirmed in 2D differentiated colonoids harvested from $Cldn23^{f/f}$ and $Cldn23^{ERΔIEC}$ mice (Fig. 3b). Interestingly, analogous to the paucity in CLDN2 expression in differentiated luminal epithelial cells at the top of the crypt–luminal axis[53], CLDN2 protein expression is not detected in differentiated colonoid monolayers from both $Cldn23^{f/f}$ and $Cldn23^{ERΔIEC}$ mice (Fig. 3b).

To examine effects of CLDN23 expression on cellular distribution of CLDN3, CLDN4, and ZO1, we analyzed localization of these proteins in co-culture experiments using primary colonoids derived from $Cldn23^{ERΔIEC}$ and $Cldn23^{f/f}$ mice. Immunofluorescence labeling and confocal microscopy revealed linear/sharp localization of CLDN23, CLDN3, and CLDN4 in the TJ plasma membrane of colonoids derived from $Cldn23^{f/f}$ mice (Fig. 3c, arrows). However, in $Cldn23^{ERΔIEC}$ derived colonoids, this linear localization was reduced and diffuse expression of CLDN3 and CLDN4 was observed in the lateral membrane and in the cytoplasm beneath the plasma membrane (Fig. 3c, arrowheads). In contrast, the TJ distribution of ZO1 was not influenced by the absence of CLDN23 expression (Fig. 3c). These observations suggest that CLDN23 modulates the targeting and/or stabilization of CLDN3 and CLDN4 at TJ plasma membrane.

To gain further insight into CLDN23-mediated regulation of CLDN3 and CLDN4 localization, we examined TJ morphology in control and CLDN23 overexpressing IECs. Immunofluorescence labeling of

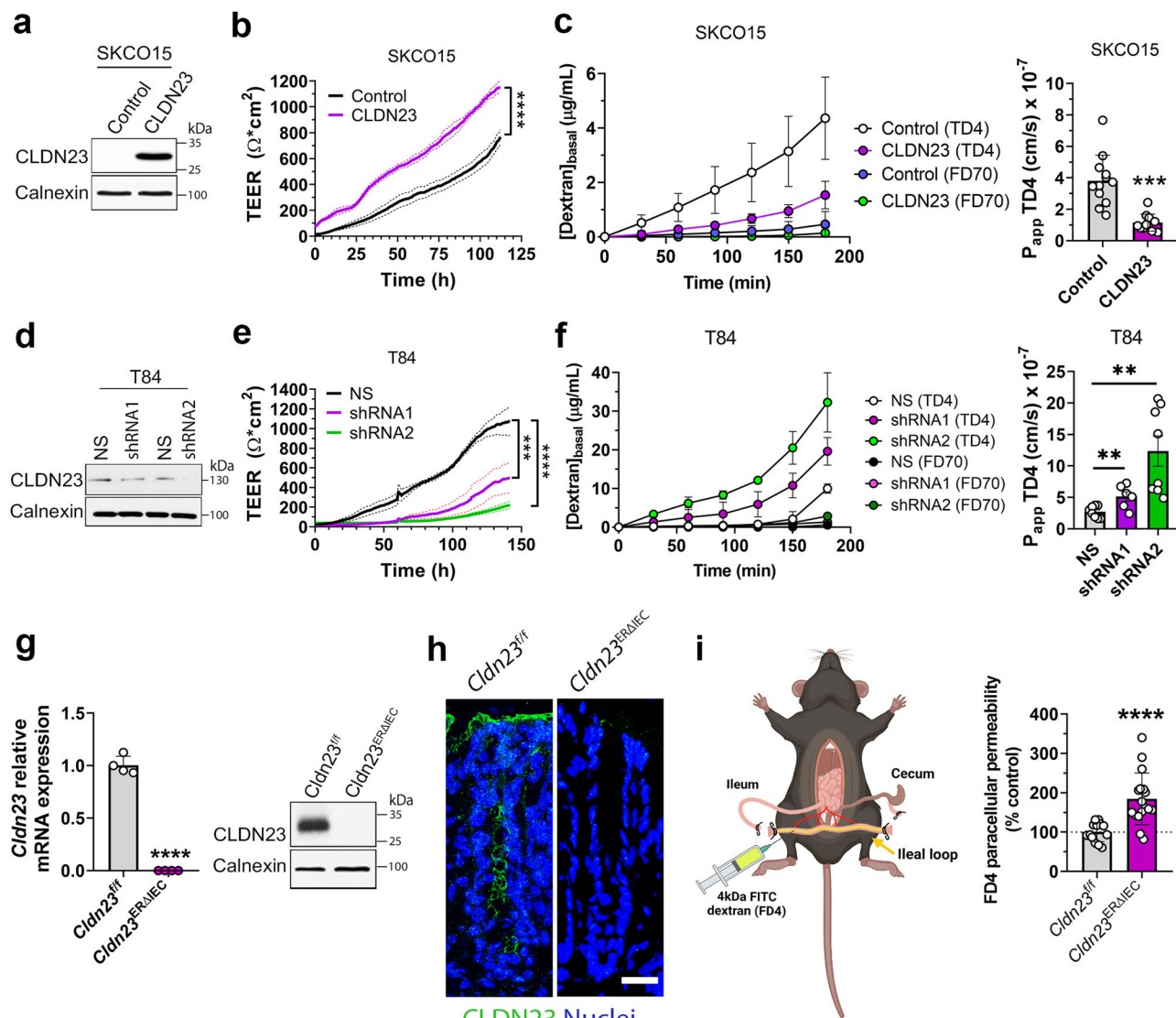

**Fig. 2 | CLDN23 regulates intestinal epithelial barrier function.**
**a** Immunoblotting for CLDN23 and Calnexin (loading control) in SKCO15 cells with ectopic expression of full-length human CLDN23 protein versus a 10 amino acid myc-tag protein (Control). **b** Representative graph showing TEER of SKCO15 cells overexpressing CLDN23 vs control monolayers was measured continuously for 5 days. Data are mean ± SD and represent three individual experiments, each with six technical replicates. ****$p \le 0.0001$; two-way ANOVA with Tukey's posttest. **c** Left, paracellular flux rate of 4 kDa TRITC-dextran (TD4) and 70 kDa FITC-dextran (FD70) across monolayers overexpressing CLDN23 and control SKCO15 cells ([Dextran]$_{basal}$). Right, rate of change of TD4 flux was utilized to calculate the apparent permeability ($P_{app}$) of each individual sample. Data are mean ± SD and are representative of three individual experiments, each with six technical replicates. ***$p = 0.001$; two-tailed Student's $t$ test with Welch's correction. **d** CLDN23 expression in T84 IECs transduced with two shRNAs against *CLDN23* were compared with scramble non-silencing shRNA control cells (NS). Immunoblot images are representative of three independent experiments. **e** Representative graph showing TEER of T84 *CLDN23* KD (shRNA1 and 2) and NS monolayers was measured continuously for 5 days. Data are mean ± SD and represent two independent experiments, each with four technical replicates per condition. ***$p \le 0.001$; ****$p \le 0.0001$; two-way ANOVA with Tukey's posttest. **f** Paracellular flux rate of

4 kDa TRITC-dextran (TD4) and 70 kDa FITC-dextran (FD70) across monolayers of T84 *CLDN23* KD (shRNA1 and 2) and NS control monolayers ([Dextran]$_{basal}$). Rate of change of TD4 flux was utilized to calculate the apparent permeability ($P_{app}$) of each individual sample. Data are mean ± SD and represent two individual experiments. Each point represents an individual cell monolayer ($n = 8$ (NS), 6 (shRNA1), and 8 (shRNA2)). **$p = 0.0068$ (NS vs shRNA1), **$p = 0.0015$ (NS vs shRNA2); two-tailed Student's $t$ test. **g** Left, *Cldn23* mRNA expression in *Cldn23*$^{ERΔIEC}$ or *Cldn23*$^{f/f}$ IECs. Points represent values from individual mice. Data are mean ± SD and represent two independent experiments, 4 mice per group. ****$p \le 0001$; two-tailed Student's $t$ test. Right, immunoblotting for CLDN23 and Calnexin (loading control) in colonic IECs from *Cldn23*$^{ERΔIEC}$ and *Cldn23*$^{f/f}$ mice. **h** Confocal images of colonic tissue sections of *Cldn23*$^{ERΔIEC}$ and *Cldn23*$^{f/f}$ mice stained for CLDN23 (green) and nuclei (DAPI/blue). Scale bar: 50μm. **i** Left, schematic of the intestinal loop model used to assess intestinal epithelial permeability to 4 kDa FITC dextran in vivo in *Cldn23*$^{ERΔIEC}$ and *Cldn23*$^{f/f}$ mice. Right, CLDN23 depletion resulted in increased intestinal permeability to 4 kDa FITC dextran in vivo. Histograms represent the mean ± SD from three independent experiments. Each point represents an individual mouse ($n = 14$ (*Cldn23*$^{f/f}$) and 16 (*Cldn23*$^{ERΔIEC}$)). ****$p < 0.0001$; two-tailed Student's $t$ test with Welch's correction.

CLDN3, CLDN4, and ZO1 in CLDN23 overexpressing IECs revealed formation of TJ spikes oriented in a perpendicular direction along cell−cell contacts (Fig. 4a, arrows). In contrast, the TJs of control cells displayed a more linear structure with fewer perpendicular spikes (Fig. 4a).

Given that changes in paracellular permeability of macromolecules are attributed to the complexity of the TJ-strand network, we next performed super-resolution stimulated emission depletion (STED) microscopy in SKCO15 IECs to determine whether CLDN23 overexpression resulted in changes in TJ strand

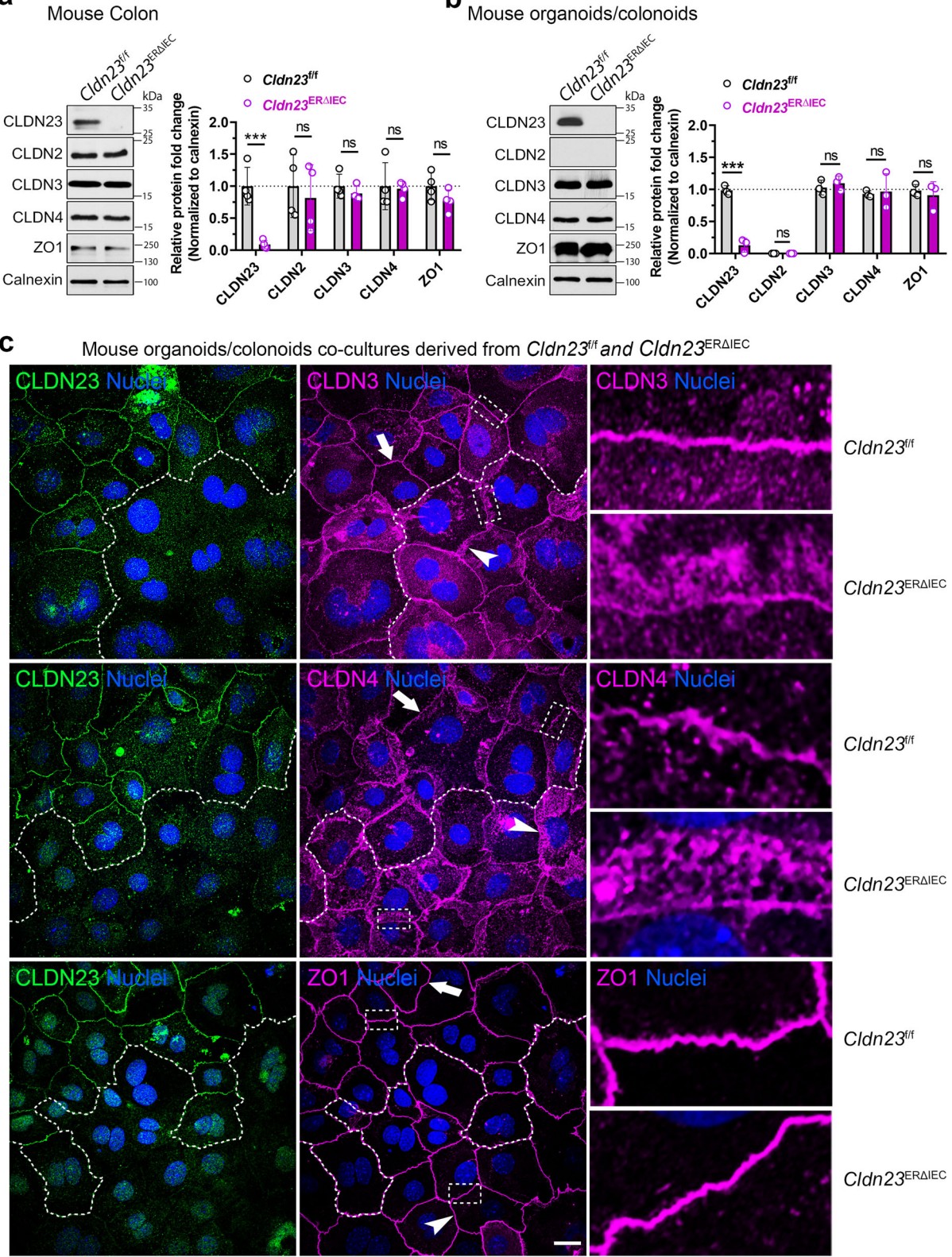

**Fig. 3 | CLDN23 stabilizes CLDN3 and CLDN4 at the TJ plasma membrane without affecting protein expression levels.** Left, representative immunoblots for CLDN23, CLDN2, CLDN3, CLDN4, ZO1 and Calnexin (loading control) in **a** whole colon and **b** murine colonoids derived from tamoxifen-treated *Cldn23*^ERΔIEC and *Cldn23*^f/f mice. Right, histograms represent the mean ± SD from three independent experiments. Each point represents an individual mouse (total of 4 per group). \*\*\**p* < 0.001; ns, not significant; **a**, **b** two-tailed Student's *t* test. **c** Representative confocal images of murine colonoid co-cultures derived from tamoxifen-treated *Cldn23*^ERΔIEC and *Cldn23*^f/f mice and stained with anti-CLDN23 (green), and either anti-CLDN3 (magenta), anti-CLDN4 (magenta), or anti-ZO1 (magenta) antibodies and DAPI (blue) as a nuclear counterstain. Dotted line indicates the border between *Cldn23*^f/f and *Cldn23*^ERΔIEC colonoids and dotted rectangles mark zoomed-in areas shown on the right. Scale bar: 20 µm.

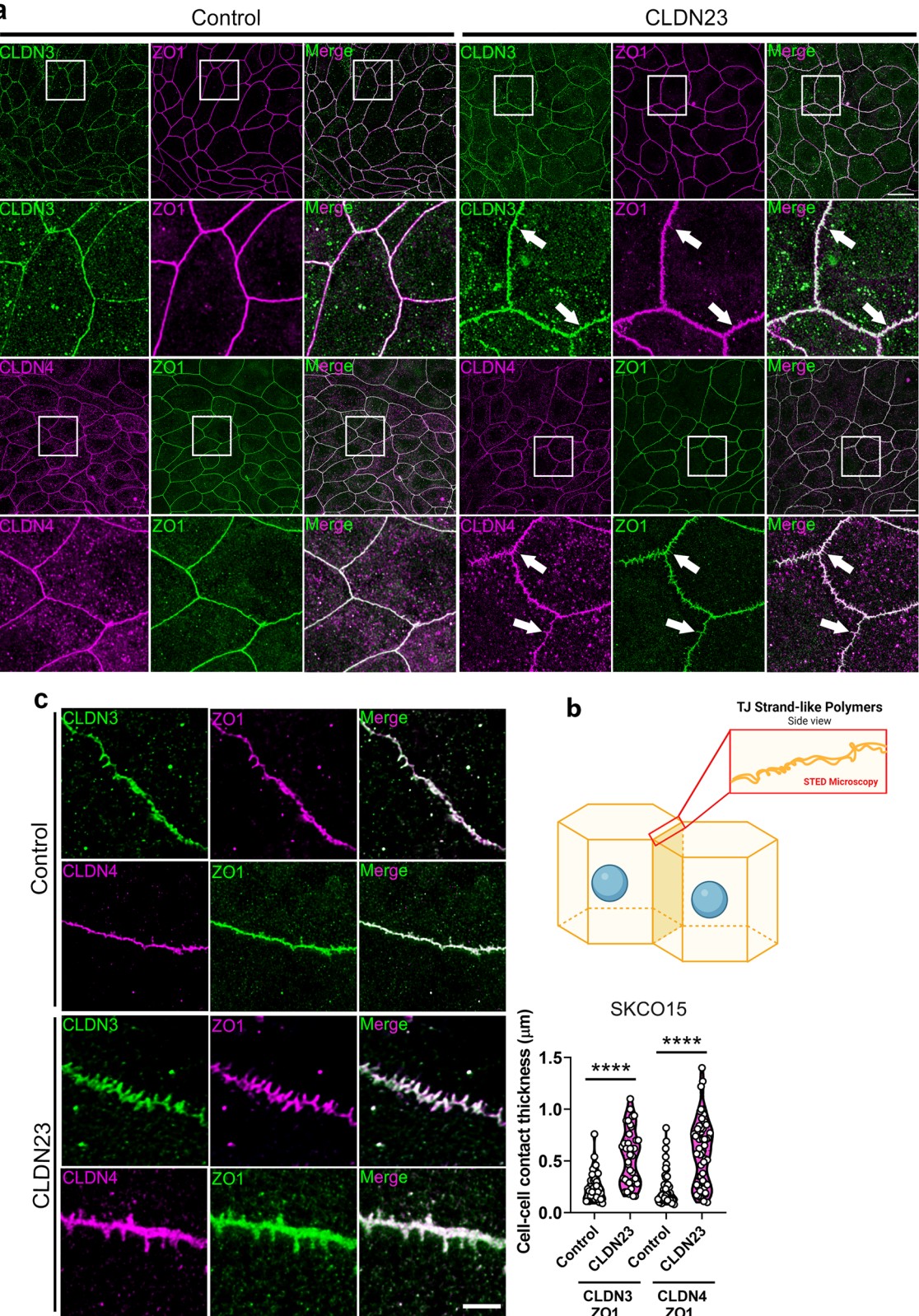

**Fig. 4 | CLDN23 influences the TJ morphology of intestinal epithelial cells.**
**a** Immunofluorescence staining and representative deconvoluted confocal images of control SKCO15 and CLDN23 overexpressing SKCO15 monolayers stained with anti-ZO1 (magenta & green) and either anti-CLDN3 (green), or anti-CLDN4 (magenta) antibodies. Scale bar: 20µm. Arrows point to TJ spike formation along cell-cell contacts. **b** Schematic representing the visualization of TJ strand formation employing super-resolution STED microscopy. Created with BioRender.com. **c** Left, representative super-resolution STED microscopy images in control SKCO15 IECs and CLDN23 overexpressing SKCO15 IEC monolayers stained with anti-ZO1 (magenta or green) and either anti-CLDN3 (green), or anti-CLDN4 (magenta) antibodies. Scale bar: 20 µm. Right, histograms showing cell–cell contact thickness. Results show the mean ± SD of two independent experiments. A total of 33 (CLDN3/ ZO1) and 50 (CLDN4/ZO1) cell-cell contacts were analyzed for control cells, while 38 (CLDN3/ZO1) and 55 (CLDN4/ZO1) were analyzed for CLDN23 overexpressing cells. ****$p < 0.0001$; statistical analysis was done with two-tailed Student's $t$ test.

architecture (Fig. 4b). While the predominant TJ morphology in control IECs was that of thin TJ strand-like polymers, CLDN23 overexpression resulted in a more complex TJ meshwork characterized by increased numbers of TJ strand-like polymers and the formation of multiple spikes resulting in the generation of broader cell–cell contacts (Fig. 4c). Taken together, these data support the importance of CLDN23 in regulating IEC barrier function through recruiting other CLDNs to the TJ and by augmenting TJ architecture complexity.

## CLDN23 interacts in *trans* with CLDN3 and CLDN4, but not CLDN2

Given our findings indicating that CLDN23 strengthens IEC barrier function and recruits CLDN3 and CLDN4 to the TJ plasma membrane, we hypothesized that CLDN23 may indirectly control epithelial barrier function through interactions with barrier-forming CLDNs. Therefore, we examined CLDN23 interactions in *trans* with CLDN3 and CLDN4 (enriched in differentiated IECs), and channel-forming CLDN2 that is expressed in crypt base IECs. We utilized HeLa cells for these assays because they provide a CLDN-null background while retaining the expression of scaffolding proteins, including ZO proteins, that are crucial for CLDN polymerization at TJs[43,70] (Supplementary Fig. 2). As shown in Fig. 5a, HeLa cells expressing CLDN2, CLDN3, CLDN4, or CLDN23 displayed membrane localization of these proteins at cell-cell contacts (arrows), suggesting homotypic *trans* interactions. To analyze CLDN23-mediated heterotypic *trans* interactions, HeLa cells expressing CLDN23 were co-cultured with HeLa cells expressing CLDN2, CLDN3, or CLDN4, and fluorescence signal colocalization measured by confocal microscopy as indicated in the schematic diagram in Fig. 5b[69]. As shown in Fig. 5c, CLDN23 colocalized with CLDN3 and CLDN4 at cell-cell contacts (arrow). In contrast minimal co-localization of CLDN23 was observed with CLDN2. Taken together, these results suggest that CLDN23 can engage in specific heterotypic *trans*-interactions with CLDN3 and CLDN4 that are notably enriched in luminal IECs (Fig. 5c).

To further determine how CLDN23 modulates epithelial barrier function, we explored whether CLDN23 preferentially interacts in *trans* with either CLDN3, CLDN4, or itself. Computational modeling between *cis* homodimers of human CLDN23 with *cis* homodimers of human CLDN2, CLDN3, CLDN4, and CLDN23 (Fig. 5d) was performed in YASARA (*details in methods section*) followed by molecular dynamics (MD) simulations to evaluate non-bonded (*trans*) interaction energies and determination of preferential association (Fig. 5e). Of note, the structures of human CLDN2, CLDN3, CLDN4, and CLDN23 were obtained by homology modeling employing crystal structure templates of claudins with >30% sequence identity to the sequence being modeled (Supplementary Fig. 7). The non-bonded interaction energy calculation includes the contributions from the Lennard Jones and Coulombic interaction energies (Supplementary Fig. 8a, b) where negative energy values are indicative of stable interactions. The *trans* heterotypic complex CLDN23/CLDN4 showed lowest energy (−1648.7 kJ/mol ± 38.8) when compared to CLDN23/CLDN3 (−1299.2 kJ/mol ± 58.3) or CLDN23/CLDN23 homodimers (−1037.3 kJ/mol ± 24.5) (Fig. 5e). The *trans* interaction between CLDN2 and CLDN23 was predicted to have the highest energy (−991.4 kJ/mol ± 26.9) among the compared heterotypic combinations, however its energy of interaction was more energetically favorable than that of the well-established homotypic interaction between CLDN3/CLDN3[44,71,72]. The Coulombic interaction energy between CLDN23 and CLDN2 resulted in a positive value (5.8 kJ/mol ± 1.9), suggesting a degree of repulsion between the interacting entities (Supplementary Fig. 8a). Taken together, these results imply a preferential association in *trans* of CLDN23 with CLDN4 and CLDN3 as opposed to a homotypic CLDN23/CLDN23 interaction.

## CLDN23 interacts with CLDN3 and CLDN4 in *cis* at the plasma membrane

We next interrogated possible *cis* interactions between CLDN23, CLDN3 and CLDN4 in IECs using a proximity ligation assay (PLA) that facilitates detection of protein-protein interactions in situ at distances <40 nm. Only antibodies that recognize CLDN tails (C-terminal region) were used for the PLA assay, therefore avoiding detection of CLDN *trans* interactions in the lateral plasma membrane of adjacent cells (Supplementary Fig. 9). As can be seen in Fig. 6a *cis* interactions between CLDN23 and CLDN3 or CLDN4 were observed (magenta fluorescent spots) primarily at cell-cell borders in murine colonoids. These observations were corroborated using SKCO15 cells expressing endogenous CLDN3 and CLDN4, which were transiently transfected with full-length human *CLDN23* cDNA (50% transfection efficiency). As shown in Supplementary Fig. 10, positive PLA signals were detected in CLDN23 overexpressing cells but not in CLDN23 non-expressing cells (asterisks), confirming PLA specificity. Furthermore, PLA signals were enriched at cell-cell contacts in CLDN23 overexpressing cells with a similar level of association between CLDN23/CLDN3 and CLDN23/CLDN4 observed (Supplementary Fig. 10).

To complement in vitro results and investigate conformational dynamics and stability of *cis* interactions between CLDN23, CLDN3 and CLDN4, we performed computational modeling using the Protein AssociatioN Energy Landscape (PANEL) approach[73]. Using the PANEL method, we were able to assess possible conformations of membrane associations between CLDN23 and CLDN3 or CLDN4. By sampling rotational orientation and non-bonded association energies, a comprehensive interaction energy landscape was obtained for each dimer. Consistent with PLA results in Fig. 6a and Supplementary Fig. 10, PANEL analyses predict that CLDN23 interacts in *cis* with CLDN3 and CLDN4 in stable low-energy conformations represented in black (Fig. 6b, left panels). Of interest, using PANEL analysis we systematically evaluated all stable dimers in each panel plot and determined if ECL domains would lead to the formation of a pore or a barrier. We identified key stable pore-forming orientations for homo- and heterodimers of CLDN3, CLDN4, and CLDN23 when each CLDN was rotated to an angle of 270 ± 10° (Fig. 6b, left panels/arrows). These configurations are represented in a 3D in silico ribbon diagram (Fig. 6b, right panels). The interaction energies between CLDN23, CLDN3, and CLDN4 revealed pore-forming homodimers as well as stable low-energy homodimers that would be unable to create pores. Energy values predict preferential interactions between CLDN4/CLDN4 in *cis* compared to CLDN3/CLDN3 and CLDN23/CLDN23 (Fig. 6d, e). Interestingly, CLDN23 is predicted to form the most stable heteromeric *cis* interactions with CLDN4, followed by CLDN3 (Fig. 6d, e). Collectively, these observations support a model in which CLDN23 preferentially engages in heteromeric *cis* and heterotypic *trans* interactions with CLDN3 and CLDN4 in order to regulate TJ pore selectivity and epithelial barrier function. In silico simulations of claudin-claudin interactions between CLDN23, CLDN3 and CLDN4 are consistent with our experimental results.

## In silico modeling suggests that CLDN23 interacting with CLDN3 and CLDN4 restricts formation of paracellular pores

To further investigate physiological consequences of CLDN23 interactions with CLDN3 and CLDN4 in regulating IEC paracellular permeability, we performed molecular docking and dynamics simulations to analyze the pore diameter resulting from the assembly of homotetrameric (CLDN23/CLDN23, CLDN3/CLDN3, CLDN4/CLDN4) versus heterotetrameric (CLDN23/CLDN3, CLDN23/CLDN4, CLDN3/CLDN4) *cis* and *trans* interactions. Pore structures corresponding to each CLDN combination involving CLDN23, CLDN2, CLDN3 and CLDN4 were modeled using CLDN15 channel forming *trans* interaction (tetramer) model as a template[74,75] (details in "Methods" section). In this model, the extracellular domains, ECS1 and ECS2, of these monomers fold into

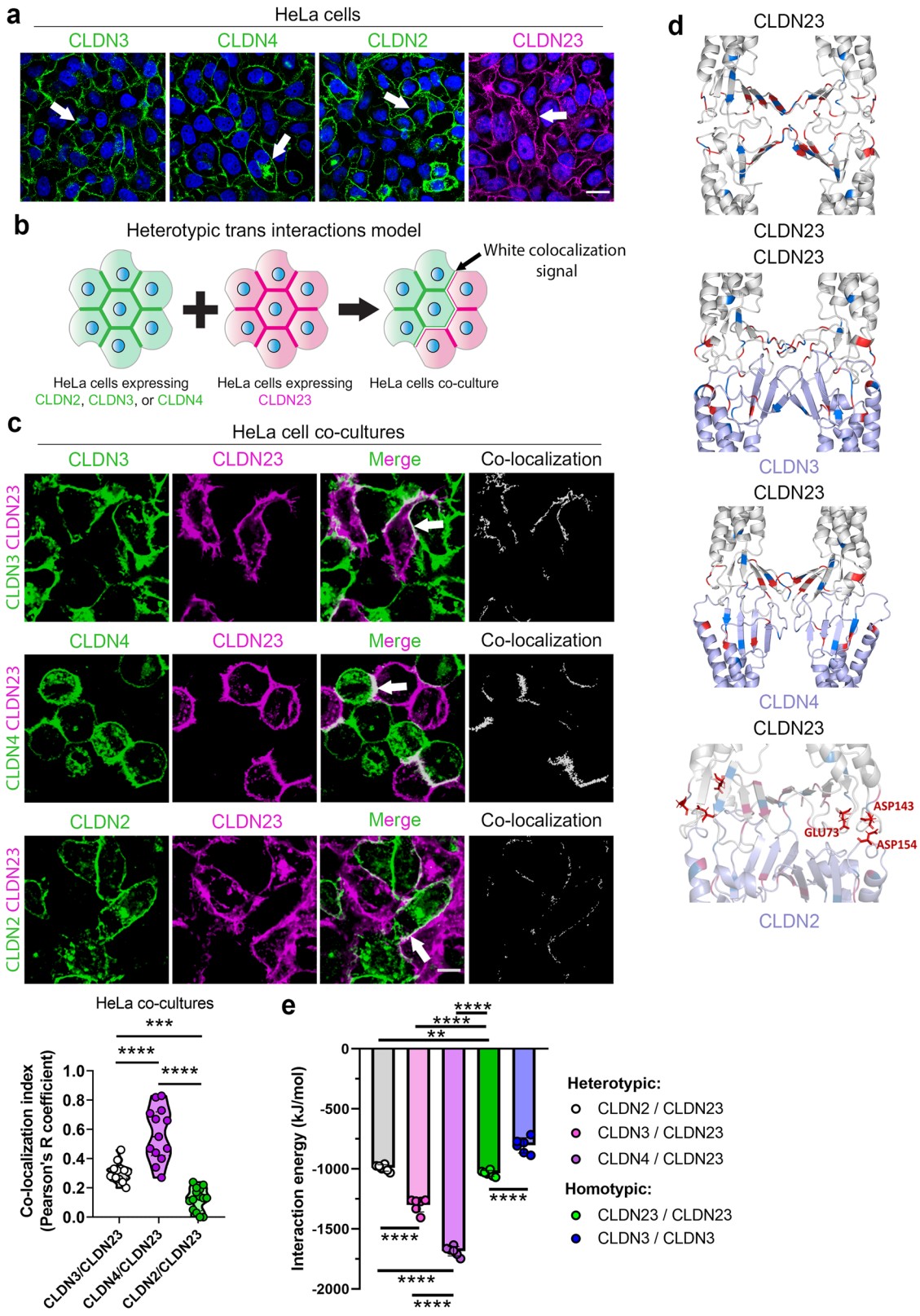

beta sheets and interact head-on (in *trans*) to create the pore structures. Resulting paracellular pores can discriminate ion transport based on solute charge and size[8,76]. As shown in Fig. 7a, homotypic *trans*-interactions between CLDN3/CLDN3 and CLDN4/CLDN4 displayed minimum pore diameters of -5.25 Å and -1.11 Å, respectively. Homotypic *trans*-interactions CLDN23/CLDN23 generated a pore diameter of 3.44 Å. *Cis* heterodimers made up of CLDN3/CLDN4

within the same plasma membrane formed homotypic *trans*-interactions with a minimum pore diameter of -5.73 Å (Fig. 7b). In the presence of CLDN23, heteromeric complexes of CLDN23/CLDN3 and CLDN23/CLDN4 within the same plasma membrane, engaged in heterotypic *trans*-interactions that resulted in smaller pore diameters (<2 Å) (Fig. 7c). Interestingly, pore formation was not observed when heteromeric CLDN23/CLDN3 and CLDN23/CLDN4 complexes were

**Fig. 5 | CLDN23 interacts in *trans* with CLDN3 and CLDN4, but not CLDN2.**
**a** Immunofluorescence staining and confocal images of HeLa cell monolayers singly expressing CLDN3 (green), CLDN4 (green), CLDN2 (green) or CLDN23 (magenta). Arrows point to homotypic *trans* interactions at cell-cell contacts. Nuclei were stained with DAPI (blue). Scale bar: 20 μm. **b** Schematic depicting HeLa cell co-culture model used to analyze heterotypic *trans* interactions between CLDN23 and either CLDN2, CLDN3, or CLDN4. **c** Top, immunofluorescence staining and confocal images of HeLa cells expressing either CLDN3 (green), CLDN4 (green), or CLDN2 (green) co-cultured with HeLa cells expressing CLDN23 (magenta). Heterotypic *trans* interactions were investigated by analyzing the colocalization index from the white signal (arrow) at cell-cell contacts. Scale bar: 20 μm. Bottom, bar graph represents the colocalization analysis, generated by Pearson's correlation coefficients, between CLDN23 and either CLDN3 ($R = 0.31$), CLDN4 ($R = 0.57$), and CLDN2 ($R = 0.12$) at cell–cell contacts. Data are mean ± SD of three independent experiments. A total of 13 (CLDN3/CLDN23), 14 (CLDN4/CLDN23), and 14 (CLDN2/CLDN23), images per condition were analyzed. ***$p \leq 0.001$, ****$p \leq 0.0001$; two-tailed Student's *t* test. **d** Claudin tetramer structures showing variable *trans* interfaces formed by CLDN23/CLDN23, CLDN23/CLDN3, CLDN23/CLDN4, CLDN23/CLDN2 interactions. The secondary structure of CLDNs is shown in ribbon representation with CLDN23 in gray and other CLDNs in light purple. *Trans* interaction structure shows proximal placement of negatively charged amino acids GLU73 and ASP143 from CLDN23 with ASP154 from CLDN2 are colored in red. **e** Bar plot showing the total interaction energy between *trans* interfaces formed by *cis* dimers of CLDN23 with *cis* dimers of CLDN2, CLDN3, CLDN4, and CLDN23. Results show the mean ± SD of five independent experiments for each tetramer. Statistical analysis was done with one-way ANOVA with Tukey's posttest. **$p \leq 0.01$; ****$p \leq 0.0001$.

engaged in homotypic *trans*-interactions (Fig. 7d). These observations are consistent with a model whereby the presence of CLDN23 in the TJ plasma membrane results in a significant reduction in pore diameter and total number of pores, supporting a role for CLDN23 in altering the pore architecture of CLDN3- and CLDN4-generated channels to regulate epithelial paracellular permeability to ions.

## CLDN23 may influence the charge selectivity of CLDN3 and CLDN4 channels

To explore the role of CLDN23 in regulating ion transport, we determined single ion permeabilities by analyzing dilution potentials in SKCO15 IECs overexpressing CLDN23 and in T84s IECs with *Cldn23* KD. Overexpression of CLDN23 led to significant decrease in both sodium ($Na^+$) and chloride ($Cl^-$) permeabilities. A modest reduction in the permeability ratio of $Na^+$ to $Cl^-$ ($P_{Na^+}/P_{Cl^-}$) was observed, indicating reduced cation selectivity (Fig. 8a and Supplementary Fig. 11). In contrast, loss of IEC CLDN23 expression resulted in significantly increased $Na^+$ and $Cl^-$ permeabilities as well as an increase in the ratio of $P_{Na^+}/P_{Cl^-}$ (Fig. 8b), further supporting our findings that CLDN23 expression reduces paracellular permeability to both $Na^+$ cations and $Cl^-$ anions. Moreover, bi-ionic potential measurements revealed that CLDN23 overexpression resulted in decreased permeability to the alkali metal cation lithium ($Li^+$), which was proportional to the observed reduction in $Na^+$ permeability (Fig. 8c). In keeping with this, knockdown of CLDN23 expression in T84 IECs resulted in increased $Li^+$ permeability that was proportional to the increase in $Na^+$ permeability (Fig. 8d). Further investigation is needed to determine if CLDN23 regulates the paracellular transport of ions other than $Na^+$, $Li^+$, and $Cl^-$.

To analyze the molecular mechanism underlying the observed ion selectivity properties of the CLDN23-containing channels, we examined the pore lining residues of the proposed channels using computational modeling. As shown in the Fig. 8e, pores formed by homomeric/homotypic channels of both CLDN3 and CLDN4 proteins have multiple positively charged residues lining the pore center, which supports their role as cation-barriers as has been previously demonstrated[15,21]. Specifically, the CLDN3 pore displayed a net positive charge +4 mainly localized at the center of the pore where there are four lysine residues (K64). At the mouth of the pore, a net neutral charge is observed as equivalent amounts of positive and negative amino acid residues (R144 and D145) are present (Fig. 8e). The predicted pore-lining residues for CLDN4 homotetrameric channels have equal numbers of positive and negative residues (K65, D68, D146, and R158) and therefore render the pore net neutral. Interestingly, residue R81 was not present within the pore structure, however a role for this residue in mediating pore selectivity cannot be excluded (Fig. 8e). The analysis of the CLDN23 homotetrameric channel revealed a net neutral pore center containing the same amount of positively (R54 and R59) and negatively (E62 and D67) charged residues. Interestingly, the mouth of the CLDN23 pore on both sides contains a high density of D143 negatively charged aspartic acid residues (Fig. 8e). Analysis of the

pore-lining residues of the CLDN3/CLDN4 heteromeric-homotypic configuration revealed a net positive charge of +2. Interestingly, although the density of charged residues on CLDN3 is higher, the net charge at the entrance of the pore is neutral. The CLDN4 pore contains negatively charged residue R158 (Fig. 8f). Importantly, the net charge along the pore of the heteromeric heterotypic channels for CLDN3/CLDN23 and CLDN4/CLDN23 was neutral (Fig. 8g). In the heteromeric homotypic conformation of CLDN23/CLDN3 there are numerous charged amino acids along the entrance of the pore on both sides, however the pore is constricted to a diameter of 0 Å (Fig. 8h). Pore lining residues for the heteromeric homotypic CLDN23/CLDN4 conformation were not evaluated due to the absence of a pore structure (Fig. 8h). Altogether, our observations describe a system in which CLDN23 regulates paracellular permeability to ions and macromolecules during colonic IEC differentiation by stabilizing and influencing the architecture and net pore charge of CLDN3 and CLDN4 paracellular channels at the TJ. In addition, this study demonstrates that the formation of heteromeric claudin paracellular complexes can influence TJ permeability.

## Discussion

CLDNs are known to play a pivotal role in creating paracellular barriers or channels that regulate permeability to ions and water in epithelial and endothelial tissues. We have demonstrated that the addition of a single claudin protein, CLDN23, can significantly alter tight junction paracellular permeability and recruit prototypic barrier-forming claudins, CLDN3, and CLDN4, to influence TJ structure and barrier function. The finding is supported by rigorous examination using complementary cell biologic, in vivo, and computational modeling approaches. One conclusion of our analysis is that different CLDNs interact in multiple ways that alters channel architecture to change TJ permeability to ions and macromolecules.

Our study provides evidence that highlights the important role of an uncharacterized and non-classic claudin protein in controlling intestinal epithelial barrier function. Epithelial paracellular barrier properties in the intestine are regulated by the expression of different CLDNs along the crypt–luminal axis[51]. This is consistent with our previous study demonstrating that *Cldn23* mRNA expression is elevated in epithelial cells at the luminal surface of murine colonic crypts. A surface enhanced expression pattern for CLDN23 parallels that of CLDN3 and CLDN4 and sharply contrasts with that of channel-forming claudins (CLDN2 and CLDN15) which are enriched at the crypt-base[53,77]. Overall, our study provides additional insight into the mechanisms that regulate intestinal permeability along the crypt-luminal axis and identifies CLDN23 as a contributor to gut permeability.

We performed molecular dynamics simulations that suggest that heteromeric and heterotypic CLDN interactions have the potential to influence pore architecture and overall net charge of pores with important functional consequences for ion permeability. Of note,

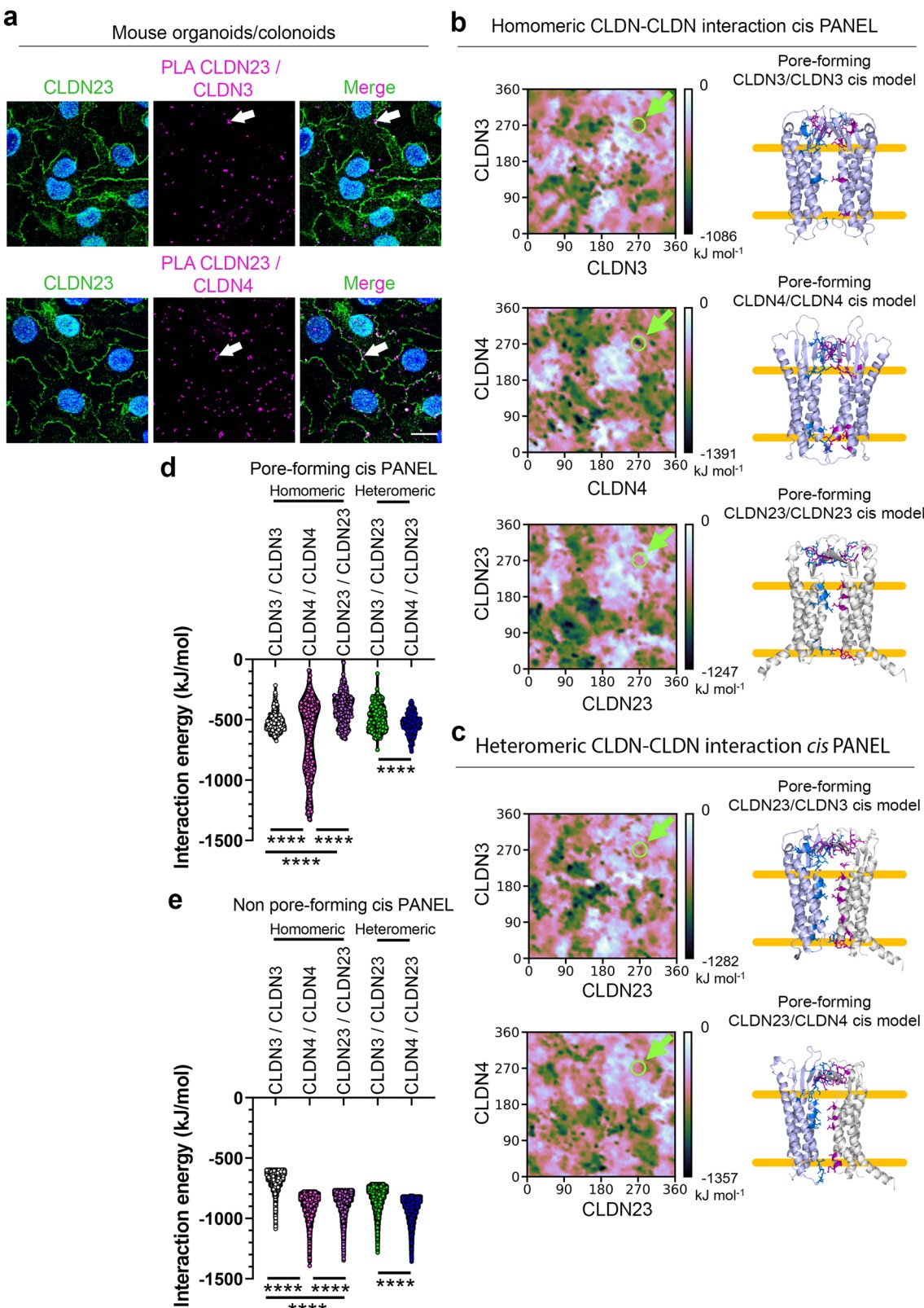

these simulations suggest that pore properties are not just regulated by claudin stoichiometry, since differently organized complexes with the same stoichiometry form pores with unique architecture. Furthermore, analysis of claudin heteromeric compatibility was validated using a computational model (PANEL) which calculates the free energy of all possible protein–protein interactions in a membrane bilayer. Computational modeling by PANEL has general applicability beyond

claudins and we anticipate that it will be a powerful approach to identify classes of favorable interactions between other classes of transmembrane proteins.

While several studies have interrogated the expression and possible contributions of CLDN23 to the neoplastic transformation of gastric, pancreatic, and colorectal cancers[57–59], ours examines a functional role for CLDN23 in regulating epithelial barrier function. The

**Fig. 6 | CLDN23 interacts with CLDN3 and CLDN4 in *cis* at the cell membrane.**
**a** Left panels, representative confocal images of mouse colonoid monolayers expressing CLDN23 at the plasma membrane. Nuclei stained with DAPI (blue) Middle panels show positive PLA signal (magenta dots) at the cell-cell contact (arrow) between CLDN23 with either CLDN3 or CLDN4 in mouse colonoids. Right panels, show merged images. Scale bars: 20 μm. **b**, **c** Left panel, interaction energy landscapes obtained from PANEL method for **b** homomeric and **c** heteromeric interactions of CLDN3 and CLDN4 with CLDN23 in which the known pore-forming rotational orientation was indicated at 270° ± 10 for each CLDN (green arrow).

Right panel, representative in silico ribbon diagrams of pore-forming homomeric (**b**) and heteromeric (**c**) interactions of CLDN3 (light purple), CLDN4 (light purple), and CLDN23 (gray). **d** Bar plot showing a comparison of energy values of pore-forming homodimers and heterodimers of CLDN3, CLDN4, and CLDN23. Results show the mean ± SD of 400 data points corresponding to the known pore-forming rotational orientation on the landscape. ****$p \le 0.0001$; one-way ANOVA with Tukey's posttest. **e** Bar plot showing energy values of non-pore-forming rotational orientations. Results show the mean ± SD of 12,960 data points. ****$p \le 0.0001$; one-way ANOVA with Tukey's posttest.

regulation of epithelial barrier function is in part attributed to the association of CLDN23 with CLDN3 and CLDN4, as CLDN23 induced localization of these claudins to TJs and away from non-junctional areas of the lateral plasma membrane. Newly synthesized CLDN4 has been previously shown to localize to the lateral membrane and slowly incorporate into the CLDN polymer network at the TJ, albeit by an unknown mechanism[30]. For CLDN3, diffuse basolateral membrane localization has been observed in murine mammary epithelial cells with simultaneous knockout of ZO-1 and knockdown of ZO-2, but not in cells that lacked only one of the ZO proteins[70]. Given that we observe TJ staining of CLDN3 and CLDN4 in cells that express CLDN23, but not in CLDN23 deficient IECs, this could represent a few scenarios, including the following: (1) Newly synthesized CLDN3 and CLDN4 traffic to the lateral membrane but CLDN23 influences their proper integration into TJ strands; (2) CLDN3 and CLDN4 depolymerize from TJ strands and distribute to the lateral membrane in the absence of CLDN23; or (3) Incorporation of CLDN3 and CLDN4 in the TJ strands can be regulated by their intracellular vesicular trafficking with CLDN23 in IECs. These possibilities raise the question of whether some CLDNs (e.g., CLDN23) can act as scaffolding units to regulate the spatial recruitment and/or stabilization of CLDN monomers in TJ strands.

In addition to CLDN23 engaging in *cis* heteromeric associations with CLDN3 and CLDN4 at cell-cell junctions, we also found evidence that these claudins can associate in intracellular vesicular compartments. Given our in silico PANEL plot analysis that identified both pore-forming and non-pore-forming *cis* interactions, we speculate that intracellular *cis* associations may correspond to non-pore forming CLDN23/CLDN3 and CLDN23/CLDN4 configurations that might facilitate intracellular vesicular trafficking to the TJ, recycling, or degradation. In support of this, others have reported that CLDN4 co-localizes with CLDN8 in intracellular vesicles, suggesting that both CLDNs traffic together to the TJ[20], and that without CLDN8, CLDN4 is confined to the endoplasmic reticulum and Golgi apparatus[20]. Furthermore, it has been suggested that CLDN8 is required for proper TJ integration of CLDN4[20]. Similarly, colocalization of intracellular CLDN16 and CLDN19 has been reported in kidney epithelial cells, suggesting co-trafficking[78]. Interestingly, heteromeric CLDN16/CLDN19 interactions were mutually dependent as absence of either of these CLDNs resulted in the absence of the other in epithelial cells of the thick ascending limb[52]. Highlighting the importance of CLDN family co-expression and co-localization in the intestine; our data support that CLDN23 regulates the integration of CLDN3 and CLDN4 at TJs in IECs in order to regulate intestinal barrier function.

Using SKCO15 IECs with CLDN23 overexpression we observed that CLDN23 increased the branching and thickness of TJ strand-like polymers formed by CLDN3 and CLDN4. In contrast, loss of CLDN23 in SKCO15 resulted in CLDN3 and CLDN4 TJ strands that were thinner, less branched, and appeared to exhibit discontinuous CLDN staining. Similar to our observations, studies in epithelial-like SF7 cells transfected with single CLDNs showed that only CLDN7, CLDN14, and CLDN9, could polymerize independently into strands while others, including CLDN3 and 4, were unable to do so[79]. Similarly, a recent study in MDCK cells showed that CLDN4 is unable to form homomeric polymers despite having two highly conserved sites in its first

extracellular loop, cis-1 and X-I, which are thought to mediate claudin polymerization[22]. Therefore, it is possible that anchoring or scaffolding proteins may be required for proper integration and polymerization of claudins into TJ strands. The observation that CLDNs 1, 2, 3, 5, and 12 can form heteropolymers[80], alongside our data identifying that CLDN23 can associate in *cis* with CLDN3 and CLDN4, further suggests that specific CLDNs can act as integral membrane scaffolding units that allow other CLDNs to polymerize into TJ strands. In contrast, we and others have also suggested that certain CLDNs may act to disrupt TJ strands formed by other CLDN family members. Specifically, CLDN4 mediated disruption of TJ strands formed by CLDN2 and 15 has been demonstrated[22,47]. However, taken together, our current findings support a role for CLDN23 acting as an integral membrane scaffolding unit that promotes CLDN3 and CLDN4 mediated TJ strand polymerization and complexity. Furthermore, these data suggest that specific CLDNs can serve as positive or negative regulators of other CLDN family members by controlling their TJ integration, endocytosis and strand structure forming properties.

Closer examination of CLDN23-mediated effects on strand architecture revealed the presence of interesting subdomains that appeared as "spikes" on STED microscopy imaging. In the literature, TJ spikes have been defined as asymmetric deviations from linear TJ morphology that appear as projections that orient in a perpendicular direction from junctions[81]. While some studies have correlated the presence of TJ spikes with increased tissue paracellular permeability (or "leakiness"), such "spikes" were shown to project from intact regions of the TJ that did not represent sites of paracellular leak[81,82]. These studies have also suggested that TJ spikes serve as an active location of vesicle fusion and budding[82]. In our model epithelial cell lines, the appearance of CLDN23-mediated TJ strand spikes correlated with higher TEER values and lower TD4 permeability, supporting the idea that junctional spikes form from mature TJs. Furthermore, it is tempting to speculate that these structures might represent areas of enhanced TJ strand reinforcement that prevent strand discontinuities that would result in increased permeability to macromolecules. Another interesting possibility is that CLDN23 maintains a claudin "reserve" near the TJ to allow for fast replacement of claudins in the setting of junctional strand breaks.

Formation of functional TJ channels requires paired TJ strands, in which each strand on adjacent cells tightly associates across the cell membranes[83]. As such, TJ strand pairing requires *trans* interactions between CLDNs that may be either homotypic or heterotypic. To date, heterotypic CLDN-CLDN interactions reported in the literature include CLDN3 association with CLDN1, CLDN2, and CLDN5[43,45,46]. In this study, we report that similar to CLDN3, CLDN23 exhibits versatile binding interactions with CLDN3 and CLDN4 in *trans* (heterotypic interactions) and in *cis* (heteromeric interactions). Using a reductionist approach, cocultures of HeLa cell clones that express a single claudin allowed us to experimentally determine that CLDN23 favors heterotypic associations with CLDN3 and CLDN4 over CLDN2. Computational modeling analyses of *trans* association energies validated these observations and suggest that heterotypic associations of CLDN23 with CLDN3 or CLDN4 are energetically favored over homotypic CLDN23 interactions. To our surprise, homotypic CLDN3 associations, which have been experimentally validated in immortalized kidney epithelial cells[44,72,84],

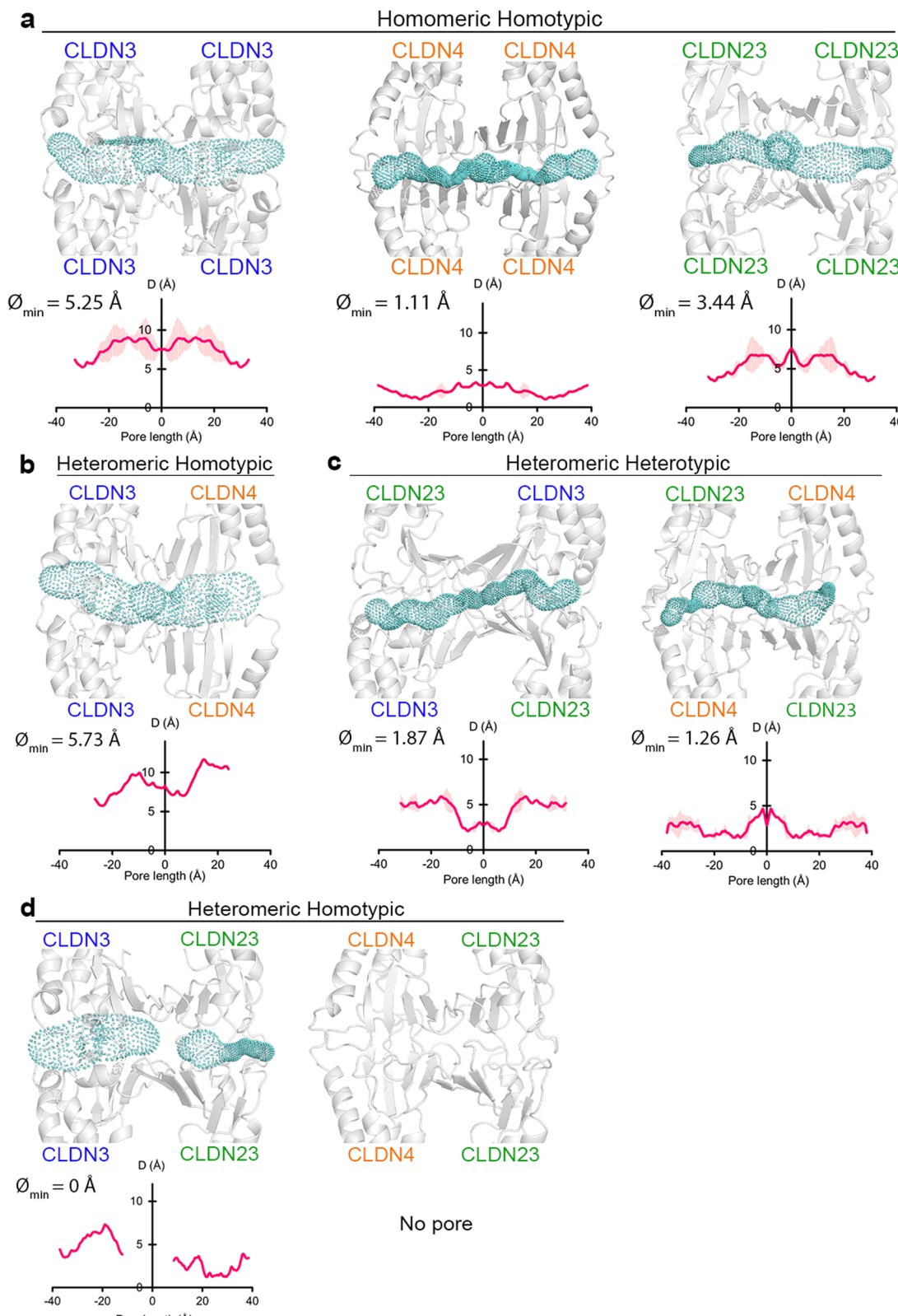

**Fig. 7 | CLDN23 interaction with CLDN3 and CLDN4 may restrict and block formation of paracellular pores.** CAVER analysis of CLDN tetrameric channels in all-atom resolution. The secondary structure of CLDNs is shown in ribbon representation. The pore profile (cyan) represents the available pore for ion/water transport across the tetrameric structure. Pore diameter along the length of the pore is shown in the graph below each tetramer. **a** Homomeric homotypic structures of CLDN3 (blue), CLDN4 (orange), and CLDN23 (green) as well as **b** heteromeric homotypic CLDN3 and CLDN4. **c** Heteromeric heterotypic and **d** heteromeric homotypic *trans* interaction structures formed by heterodimers of CLDN3 and CLDN4 with CLDN23.

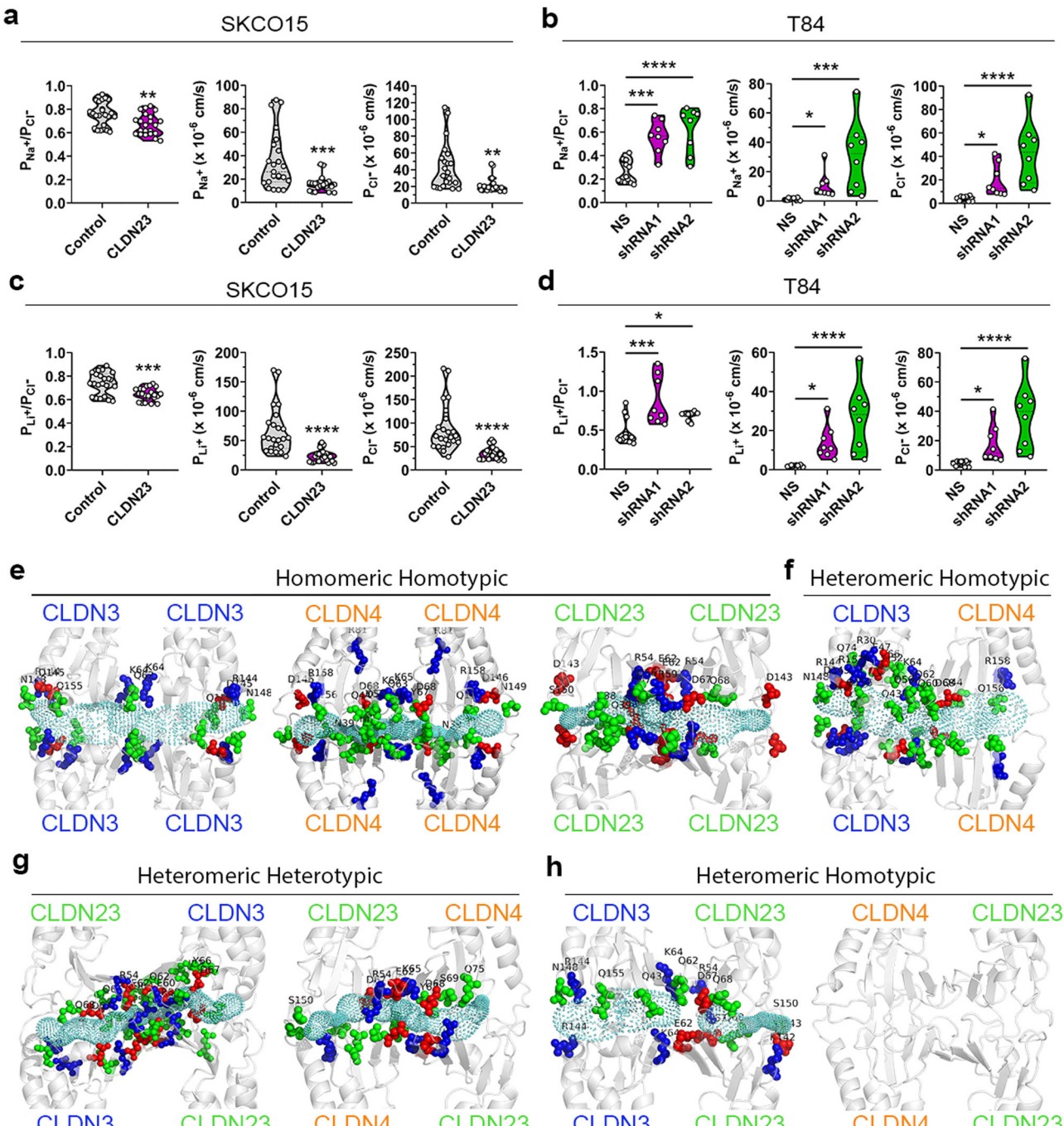

**Fig. 8 | CLDN23 may decrease the paracellular permeability of ions of either charge by influencing the charge selectivity of CLDN3 and CLDN4 pores.**
**a**, **b** Charge selectivity (ratio of permeability of $P_{Na}^+$ to $P_{Cl}$; $P_{Na}^+/P_{Cl}^-$) and individual $P_{Na}^+$ and $P_{Cl}^-$ in **a** control and CLDN23 overexpressing SKCO15 cells and in **b** T84 IECs transduced with two shRNAs against *CLDN23* compared with scramble non-silencing shRNA control cells. Data are mean ± SD and represent **a** four and three **b** independent experiments. Each dot represents an individual cell monolayer ($n$ = 22 (**a**) and 11 (**b**)). *$p$ = 0.0198 ($P_{Na}^+$ of T84 NS vs shRNA1), *$p$ = 0.0173 ($P_{Cl}^-$ of T84 NS vs shRNA1), **$p$ = 0.0031 ($P_{Na}^+/P_{Cl}^-$ of SKCO15), **$p$ = 0.0010 ($P_{Cl}^-$ of SKCO15) ***$p$ = 0.005 ($P_{Na}^+$ of SKCO15), ***$p$ = 0.0001 ($P_{Na}^+/P_{Cl}^-$ of T84), ***$p$ = 0.0003 ($P_{Na}^+$ of T84 NS vs shRNA2), ****$p$ ≤ 0.0001 ($P_{Cl}^-$ of T84 NS vs shRNA2); **a** two-tailed Student's $t$ test and **b** one-way ANOVA with Tukey's posttest ($P_{Na}^+/P_{Cl}^-$ of T84) and two-tailed Student's $t$ test ($P_{Na}^+$ and $P_{Cl}^-$ of T84). **c**, **d** Charge selectivity (ratio of permeability of $P_{Li}^+$ to $P_{Cl}$; $P_{Li}^+/P_{Cl}$) and individual $P_{Li}^+$ and $P_{Cl}^-$ in **c** control and CLDN23 over-expressing SKCO15 cells and in **d** T84 IECs transduced with two shRNAs against

*CLDN23* compared with scramble non-silencing shRNA control cells. Data are mean ± SD and represent **c** four and three **d** individual experiments. Each dot represents an individual cell monolayer ($n$ = 22 (**c**) and 11 (**d**)). *$p$ = 0.0454 ($P_{Li}^+/P_{Cl}^-$ of T84 NS vs shRNA2), *$p$ = 0.0257 ($P_{Li}^+$ of T84 NS vs shRNA1), *$p$ = 0.0211 ($P_{Cl}^-$ of T84 NS vs shRNA1), ***$p$ = 0.0002 ($P_{Li}^+/P_{Cl}^-$ of SKCO15), ***$p$ = 0.0002 ($P_{Li}^+/P_{Cl}^-$ of T84 NS vs shRNA1), ****$p$ ≤ 0.0001 ($P_{Li}^+$ of SKCO15, $P_{Cl}^-$ of SKCO15, $P_{Li}^+$ of T84 NS vs shRNA2, and $P_{Cl}^-$ of T84 NS vs shRNA2); **c** two-tailed Student's $t$ test and **d** one-way ANOVA with Tukey's posttest. **e** Homomeric homotypic structures of CLDN3 (blue), CLDN4 (orange), and CLDN23 (green) as well as **f** heteromeric homotypic CLDN3 and CLDN4. **g** Heteromeric heterotypic and **h** heteromeric homotypic *trans* interaction structures formed by heterodimers of CLDN3 and CLDN4 with CLDN23. The secondary structure of CLDNs is shown in ribbon representation. The pore profile (cyan) represents the available pore for ion/water transport across the tetrameric structure. The positively charged residues are shown in blue, negatively charged residues in red and polar residues (ASN, GLN, SER, THR, TYR) in green.

were predicted to be less energetically favored than associations between CLDN23/CLDN2.

Together our data suggest a model in which CLDN23 has the potential to preferentially associate in *trans* with other claudins in the following order: CLDN4 > CLDN3 > CLDN23 > CLDN2. In the intestine, *trans* associations between CLDN23 and CLDN2 are unlikely due to their spatial separation in the crypt-luminal axis[53]. However, it is possible that these claudins may interact in a heterotypic fashion in other contexts, such as during intestinal inflammatory states that enhance CLDN2 expression in luminal IECs, or in tissues that can co-express these claudins such as the placenta[85,86]. Interestingly, no high degree of homology and sequence identity was found between extracellular segment (ECS) sequences of CLDN23 and those of CLDN3 or CLDN4. However, previous reports show heterotypic binding compatibility between CLDN1/CLDN3 and CLDN5/CLDN3 despite low ECS sequence homology[22,24]. Furthermore, CLND3 and CLDN4 are heterotypically incompatible even though the ECS domains of these two CLDNs are highly conserved[43]. Taken together, these observations suggest that homology between ECS domains is insufficient to define specificity of heterotypic claudin interactions. Alternatively, such specificity might be regulated by electrostatic interactions between the ECS1 and ECS2 domains of the claudin pair.

Our molecular docking and dynamics simulations of homotetrameric CLDN23 channels suggest that the resulting pore is on average 1.5 times narrower than that of the homotetrameric CLDN3 channel. Moreover, simulations suggest that CLDN23 has a significant impact on paracellular channel architecture when paired with CLDN3 or CLDN4. Heteromeric heterotypic associations resulted in extensive simulated pore narrowing, whereas heteromeric homotypic channel conformations led to complete obliteration of the simulated pore opening, suggesting that CLDN23 may function to reduce the total number of available ion-permeable channels within TJ strands. Our findings further suggest that the pore is not simply affected by claudin stoichiometry, since differently organized complexes with the same stoichiometry form simulated pores with unique architecture. Pore size as well as select amino acid residues within the ECS1 are important molecular determinants of ion permeation selectivity of claudin channels[8,40,76]. For example, the aspartate-65 (D65) residue in CLDN2, and aspartate-55 (D55) and glutamate-64 (E64) residues in CLDN15 and CLDN10b determine cation selectivity[9,11,13,76,87]. Interestingly, a recent study by Hempel et al. performed amino acid substitutions to neutralize functionally important charged residues of CLDN15, aspartate to asparagine (D55N) and glutamate to glutamine (E64Q) and observed decreased ion permeabilities and a narrowing in the pore diameter[50]. This observation suggests that charged residues within the pore lining may regulate pore permeability, likely through exertion of repulsive forces among amino acids of the same charge combined with differences in pore diameter. In line with this, CLDN3 and CLDN4 channels have a positively charged residue at lysine-63 (K63) and lysine-65 (K65), respectively. However, the difference in pore diameter between these homotetrameric channels might be due to the net positive charge of +3 along the pore length of the CLDN3 channel, while the pore of the CLDN4 channel is net neutral. It is therefore conceivable that the strong positive charge influence along the CLDN3 pore creates a repulsive force that results in a larger pore diameter, whereas the net neutrality of CLDN4 allows for a smaller pore. Additionally, we observed a small simulated pore diameter for CLDN23 channel, which correlates with the presence of uncharged amino acid residues in the corresponding ion bindings sites, glycine-63 (G63) and glutamine-64 (Q64), and a neutral pore center. Therefore, it is conceivable that the neutral amino acids lining the pore of CLDN23 channels may be important for mediating pore size narrowing and may also affect the charge selectivity of the resulting ion channels.

Regarding the CLDN4 channel, previous studies have shown that arginine-81 (R81) and K65 influence ion transport across the pore[20]; however, our models and those of Berselli et al.[42] suggest that this residue is not directly lining the pore. One possibility is that R81 might influence the ion selectivity of the resulting pores by altering the structural folding aspect rather than direct involvement. Another possible scenario is that the R81 residue may function to decrease the influence of the negatively charged aspartic acid-68 (D68) and in turn allow a stronger influence of K65 on ion selectivity, supporting the role for CLDN4 as a chloride channel[20]. However, in our model the pore diameter of CLDN4 is constricted, suggesting that it could also restrict ion permeation. The idea that CLDN4 may act as both barrier- and channel-forming CLDN might be explained by a dynamic "breathing" of the pore diameter which might render CLDN4 sporadically permeable to Cl- as well as completely impermeable at times.

Here we report that expression of CLDN23 was sufficient to induce a reduction in IEC paracellular permeability to ions of either charge, specifically lithium (Li+), sodium (Na+), and chloride (Cl−) ions. These CLDN23-mediated permeability changes are consistent with those that have been published for the anion and cation-barrier claudin, CLDN3[15]. However, a more extensive analysis is needed to determine the impact of CLDN23 on paracellular permeability to other ions. Because we observed that CLDN23 showed decreased permeability to Na+ compared to the smaller Li+ cation, our findings suggest that CLDN23-containing channels likely conform to ionic sorting following the Eisenman sequence XI (Li+ > Na+ > K+ > Rb+ > Cs+) in which the ion with the smallest non-hydrated radius has a higher permeability. However, we cannot exclude the possibility that the CLDN23 channel could also conform to the non-Eisenman sequence X (Li+ > Na+ > Rb+ > K+ > Cs+)[88]. Given the relative abundance of channels that conform to the Eisenman sequence XI versus the non-Eisenman sequence X in vivo (61.40% vs. 1.04%, respectively)[88], it is likely that the CLDN23 channel follows sequence XI. Furthermore, given that Li+ has a larger solvation shell than Na+, it is tempting to speculate that the selectivity of Li+ > Na+ transport may be due to a possible dehydration effect exerted by the aspartic acid residues (D143) lining the entrance of the CLDN23 pore. To date, there have been no other reports showing direct effects of specific CLDNs on the paracellular channel architecture of other CLDNs resulting in functional effects on paracellular permeability. Further studies to assess whether other CLDN family members apart from CLDN23 can induce pore-size narrowing to affect paracellular permeability of solutes will be fundamental in advancing our understanding of TJ structure and function.

Taken together, we show a role for CLDN23 in intestinal epithelial barrier function regulation through a proposed mechanism of paracellular pore restriction and TJ strand remodeling. We believe that such CLDN23-mediated regulation of barrier function could provide physiological advantages when intestinal tissues are exposed to harmful antigens and toxins. Tightening of the epithelial barrier through formation of reinforced TJ strands and obliteration of paracellular pores may represent an important immunological defense mechanism for intestinal luminal cells that are closely exposed to the antigen-rich luminal environment. Importantly, given that we did not observe spontaneous colitis following acute CLDN23 downregulation in vivo, it is possible that the lack of IEC CLDN23 expression may come into play in the presence of a "second hit." Previous studies have shown that unchallenged mice harboring IEC-specific *Cldn2* overexpression as well as knockout of Desmocollin-2, Junctional Adhesion Molecule-A, or non-muscle myosin IIA displayed increased mucosal permeability without the development of colitis due to adaptive protective mechanisms[77,89–91]. Therefore, further studies to elucidate these and other possible non-canonical roles of CLDN23 should be performed.

An in-depth understanding of mechanisms by which specific members of the CLDN family work together to regulate paracellular pores can have important implications on therapeutic approaches to either enhance mucosal barrier function in human disease or to

increase TJ permeability for delivery of therapeutic agents. Our data suggesting that CLDN23 interacts with CLDN3 and CLDN4 to impact pore geometry and ion permeability is also applicable to other combinations of claudins. Modeling interactions between other claudins will help determine how different claudins interact, leading to testable models and new approaches to specifically fine tune TJ permeability by distinguishing structural determinants of pore formation and identifying key determinants that regulate claudin–claudin interactions.

## Methods

Our research complies with all relevant ethical regulations (IACUC and IBC) and University of Michigan School of Medicine that approved the study protocols (PRO00009903).

### Animal experiments

Mice selectively deficient in CLDN23 in the intestinal epithelium were generated by breeding *Cldn23* "floxed" (*Cldn23*^f/f) mice with mice expressing the inducible mutated estrogen receptor fused to Cre-recombinase under control of the *Villin* promoter (*Cldn23*^ERΔIEC). Six- to eight-week-old *Cldn23*^ERΔIEC and control *Cldn23*^f/f were injected intraperitoneally with 1 mg/100 μl of tamoxifen (Sigma, Cat. T5648) dissolved in 10% ethanol and sterile corn oil (Sigma, Cat. C8267) for 5 consecutive days. A similar number of female and male mice were used indistinctly for all the experiments. Animals were used 21 days after the last tamoxifen injection. Mice were kept under strict specific pathogen-free conditions with ad libitum access to normal chow and water. All experiments were approved and conducted in accordance with the guidelines set by the University of Michigan Institutional Animal Care and Use Committee.

### Antibodies

The following primary monoclonal and polyclonal antibodies were used to detect proteins by immunofluorescence (IF) or immunoblot (IB). From rabbit: anti-human/mouse CLDN23 (IB: 1/1,000; IF: 1/100) was generated; anti-human/mouse CLDN3 (Sigma, Cat. 218317, IB:1/1000; IF:1/100); anti-human CDX2 (Cell Signaling, Cat. 39775, IB: 1/1000); and anti-calnexin (Cat. PA5-34665; IB: 1/20,000). From mouse: anti-mouse CLDN2 (Invitrogen, Cat. 32-5600; IB: 1/1000; IF: 1/250); anti-human CLDN3 (Sigma, Cat. SAB4200758, IB:1/1,000; IF:1/100); and anti-human CLDN4 (Invitrogen, Cat. 32-9400, IB:1/2000; IF:1:200); anti-human/mouse ZO-1 (Thermofisher, Cat. 33-9100, IB: 1:1000, IF 1:100).

### Cell culture and TEER analysis

SKCO15 were provided and authenticated by Dr. Rodriguez-Boulan E. T84 cells were obtained from ATCC (CCL-248). SKCO15 and T84 human model IEC were cultured either on Transwell permeable supports (0.4 μm pore-size, Corning, Cat. 3460) or on tissue-culture treated plastic as previously described[92,93]. Human SKCO15 control and transformed IEC lines overexpressing CLDN23 were grown in high glucose (4.5 g/l) DMEM supplemented with 10% fetal bovine serum. Control and *CLDN23* knockdown (KD) T84s were grown in DMEM/F12 with 5% NCS. KD of CLDN23 was established by RNA interference using two different short-hairpin RNA (shRNA) against human CLDN23 or scrambled non-silencing (NS) control cloned into the pSMART vector (Dharmacon). In short, shRNA transduction (MOI of 2) of IECs was performed in 60–70% confluent IECs using spinfection protocol (1200 × *g* for 30 min at RT). Puromycin selection (6 μg/ml) was used to ensure proper KD (3 subcultures) and maintained for the entire experiment (4 μg/ml). Cells were grown for two days to reach confluency and harvested 3–4 days later to perform experiments. OE of CLDN23 was achieved by lentiviral transduction of human CLDN23 cloned into the pLex-MCS vector in 30–40% confluent IECs using spinfection. Cells were puromycin selected during two subcultures and maintained at the same concentration (2 μg/ml) to ensure proper

OE. The cells were grown for two days to reach cell confluency and harvested 3–4 days later for the experiments. SKCO15 cells expressing a 10 amino acid myc-tag protein (Control) and T84 cells transduced with a scrambled non-silencing shRNA (NS) were used as control cells. For cells cultured on Transwell permeable supports, TEER was continuously measured from each insert by using an automated cell monitoring system, cellZscope (nanoAnalytics GmbH, Munster, Germany). TEER values were obtained by using the cellZscope software, version 4.4.12.0. For dilution potential experiments, an EVOM voltmeter with an STX2 electrode (World Precision Instruments; Sarasota, FL) and Ussing chamber (Physiologic instruments) were used. Baseline resistance and transepithelial potential was subtracted from filters covered with cells and expressed as Ω·cm² and mV, respectively.

HeLa cells were obtained from ATCC (CCL-2). HeLa cells expressing CLDN2, CLDN3, CLDN4, or CLDN23 individually were grown in MEM with 10% FBS[43] and cultured in tissue-culture treated plastic or glass coverslips. HeLa cell clones were selected using either 800 μg/ml of hygromycin (HeLa CLDN2 cells) or 3 mg/ml of G418 (HeLa CLDN3, CLDN4, and CLDN23 cells) and grown in co-cultures for experiments assessing CLDN-CLDN *trans* interactions.

### Colonoid culture

To obtain 3D murine colonoids, intestinal crypts were isolated from the colon of male and female *Cldn23*^ERΔIEC and *Cldn23*^f/f mice (mice were age and sex matched between genotypes for each experiment), embedded in Matrigel (Corning, 365237, Lot 9112015), and maintained in LWRN-conditioned media supplemented with 50 ng/ml of recombinant human EGF (R&D Systems, 236-EG) and antibiotics/antimycotic (Corning, 30-003-Cl) as described previously[94]. To acutely deplete CLDN23, colonoid cultures were treated for 72 h with 1 μM (Z)-4-hydroxytamoxifen (Sigma-Aldrich, H7904) in complete media followed by passage and maintenance in Z−4-hydroxytamoxifen-free complete media. Direct 2D colonoid monolayers were generated directly from intestinal crypts as described in[95]. Isolated crypts were seeded onto collagen and laminin coated plates, Transwells, and/or cover slips. Murine 2D cultures were maintained in LWRN complete media for 24–48 h, until monolayers attained confluency, and then media was changed to differentiation media for at least 24 h to allow for epithelial differentiation. For colonoid cocultures, equal numbers of isolated crypts from *Cldn23*^ERΔIEC and *Cldn23*^f/f mice were mixed and seeded onto collagen- and laminin-coated cover slips.

### Duolink in situ proximity ligation assay

*Cis* interactions between CLDN23 and CLDN2, CLDN3, or CLDN4 were examined using the DuoLink in situ proximity ligation assay Kit (Sigma), following the manufacturer's protocol. Briefly, murine colonoids or CLDN23 OE SKCO15 cells were plated in 8-well chamber slides and cultured to 70% confluence before being fixed with 4% PFA for 15 min at 4 °C, permeabilized with a PBS+ solution containing 0.5% Tx-100 for 30 min at room temperature followed by incubation with 0.5% SDS/PBS+ solution for 10 min at room temperature. Then, the samples were blocked with 3% BSA (Sigma) for 1 h. Cells were treated with antibodies directed to CLDN23 (rabbit anti-human/mouse claudin-23; 1:100; generated in-house), and CLDN3 (mouse anti-human claudin-3; 1:200; Sigma) or CLDN4 (mouse anti-human claudin-4; 1:200; Invitrogen) overnight at 4 °C. After washing with PBS, cells were treated with PLA probes (1:5; anti-mouse PLUS, anti-rabbit MINUS) for 1 h at 37 °C in a preheated humidity chamber. After washing, ligation solution was added for 30 min at 37 °C, followed by washing and adding amplification polymerase solution for 100 min at 37 °C in a humidity chamber. The PLA generates discrete spots indicating positive interactions between two proteins, which are visualized by confocal microscopy imaging[96]. After the Duolink reaction was completed, the preparations were washed 3 times, and CLDN23 expressed at the cell borders were stained with anti-rabbit antibody coupled to Alexa Fluor 488

(Invitrogen). Slides were washed and dried before adding mounting medium and placing coverslip. Fluorescence images were acquired with a Nikon A1 confocal inverted laser microscope at the University of Michigan Biomedical Research-Microscopy Core.

## Electrophysiological measurements

xMurine colonoids, SKCO15 and T84 human model IECs were cultured on Transwell permeable supports (0.4 μm pore-size, Corning, Cat. 3460). Transepithelial electrical resistance (TEER) was measured daily using an EVOM voltmeter with an ENDOHM-12 (World Precision Instruments; Sarasota, FL). For continuous monitoring, resistance readouts were obtained using the automated cellZscope 2 system (nanoAnalytics). Ohmic resistance values were corrected for the area of Transwell as well as for the related value of a blank and reported as $\Omega \cdot cm^2$. For dilution potential measurements, an EVOM voltmeter was used. To measure voltage potential (mV), the current was clamped at 10 mA, and the electrode was allowed to calibrate in a pre-warmed Ringer's saline solution (140 mM NaCl, 2 mM CaCl₂, 1 mM MgCl₂, 10 mM glucose, and 10 mM HEPES; pH 7.3) for 30 min or until the voltage reading remained unchanged for 10 min. Measurement of ion permeabilities was performed as described in detail elsewhere and adapted to cultured monolayers[97]. All measurements were carried out in circulating Ringer's solution, gassed with 95% O₂/5% CO2 at a temperature of 37 °C and a pH of 7.35. For dilution potential measurements, the chamber system was filled with 4 ml Ringer's solution (140 mM NaCl, 2 mM CaCl₂·2H₂O, 1 mM MgCl₂·H₂O, 10 mM glucose, 10 mM HEPES and adjusted to pH 7.3 with 5 M NaOH and filtered 0.2 μm) on each side. After acclimatization of cells, 4 ml of the basolateral bathing solution were replaced by a modified Ringer's solution containing a 1:4 Ringers Saline to Ringer Mannitol (280 mM mannitol instead of 140 mM NaCl). Transepithelial resistance and voltage were recorded during the whole experiment, and permeability ratios for Na⁺ and Cl⁻ ($P_{Na^+}/P_{Cl^-}$) were calculated according to the Goldman−Hodgkin−Katz equation as described in ref. [13]. Permeabilities for Li+ were investigated by replacing 4 ml of the standard bathing solution by a modified Ringer's solution containing the 140 mM Li⁺ instead of 140 mM Na⁺. Absolute permeabilities were calculated from relative permeabilities and transepithelial resistances as described elsewhere[13].

## Intestinal Swiss rolls and tissue processing

Murine colons were harvested from *Cldn23*[ERΔIEC] and *Cldn23*[f/f] mice and prepared as Swiss rolls, as previously described[98]. Samples were then fixed in a 10% neutral buffered formalin solution overnight. After fixation, tissue was washed in PBS and 70% ethanol was added for 24 h prior to the paraffin embedding process. Tissue sections were then stained with hematoxylin and eosin (H&E) using standard protocols. Stained sections were then scanned utilizing an Aperio AT2, High Volume, Digital Whole Slide Scanning Imager (Leica Biosystems) at the University of Michigan, Department of Pathology. Images were analyzed utilizing ImageScope by Aperio, Version 12.3.3.5039.

## Intestinal epithelial differentiation assay

Epithelial cell differentiation was performed by employing a protocol previously described by Farkas et al.[63]. Colonic epithelial cell lines were plated on permeable supports and epithelial differentiation was monitored for nine days as cells established cell contacts, intercellular junctions matured, and cells differentiated into polarized epithelial cells. Then, the expression of CLDN2, CLDN3, CLDN4, and CLDN23 was analyzed by immunoblot (IB) and qPCR. CDX2 and Calnexin were used as differentiation and housekeeping markers, respectively.

## Flux measurements

Paracellular permeability was done as previously described[69]. 4-kDa fluorescein isothiocyanate-labeled (FITC)-dextran (70 kDa; 200 μg/ml) (Sigma, Cat. FD4) and tetramethylrhodamine isothiocyanate (TRITC)-dextran (4 kDa, 200 μg/ml) (Sigma Aldrich, Cat. T1037) was assessed in control, Cldn23 OE, and KD confluent monolayers grown on Transwell filters (0.4 μm pore-size filters, Corning, Cat. 3460). After TEER measurement, upper and lower Transwell compartments were washed twice with calcium-containing PBS and placed in pyruvate buffer (10 mM HEPES, pH 7.4, 1 mM sodium pyruvate, 10 mM glucose, 3 mM CaCl₂, and 145 mM NaCl) for 1 h at 37 °C. A freshly prepared solution of 4 kDa TRITC-dextran and 70 kDa FITC-dextran dissolved in pyruvate buffer was added to the top chamber of the Transwells and incubated for 3 h at 37 °C. Samples from the bottom chamber of the Transwells were collected every 30 min, and the fluorescence intensity was measured with a fluorescent plate reader. For TRITC-dextran excitation was achieved at $555 \pm 10$ nm and emission detected at $580 \pm 20$ nm. For FITC-dextran excitation was achieved at $490 \pm 10$ nm and emission was recorded at $525 \pm 20$ nm. The apparent permeability ($P_{app}$) was determined by calculating the rate of change of the paracellular flux of TD4 and FD70 every 30 min for 3 h per sample.

## Intestinal loop model

In vivo intestinal epithelial permeability at baseline was measured as previously described using an ileal loop model[67,69]. Animals were anesthetized with isoflurane (Fluriso, VETONE) at a constant rate using a rodent anesthesia vaporizer machine (E-Z Anesthesia 7000) and placed on a controlled temperature heat pad to avoid hypothermia. After disinfection of the abdominal skin, laparotomy was performed by midline incision. A 4-cm length of terminal ileum was exteriorized without rupturing of the blood supply. The loop was gently flushed with warm HBSS plus calcium and magnesium (HBSS plus; Corning Cellgro) to remove fecal contents and facilitate normalization of the volume of contents to allow for comparative analyses between groups. The four generated cut-ends were closed by ligations using non-absorbable silk suture 3.0 (Braintree Scientific). The loop was injected with 200 μl (1 mg/ml FITC labeled Dextran [4 kDa] dissolved in HBSS+) using the insertion of a 0.5″, 27-gauge needle. The loop was reinserted in the abdominal cavity; then, the peritoneum and skin were closed. After 2 h, blood was collected by cardiac puncture prior to euthanasia of the animals by cervical dislocation. FITC-dextran flux was determined by measuring plasma at 488 nm in a microplate spectrophotometer (Epoch Biotek, Vermont) and Gen5 software.

## Immunofluorescence microscopy and co-localization analysis in HeLa cells

Cells were grown on glass coverslips and fixed with 4% paraformaldehyde in phosphate-buffered saline (PBS) for 30 min at 4 °C followed by permeabilization with 0.5% Triton X-100 for 15 min and 0.5% SDS 10 min at room temperature, respectively. Cellular epitopes were blocked with 3% bovine serum albumin (BSA; Sigma-Aldrich) for 1 h at room temperature. Cells were incubated overnight at 4 °C with primary antibodies diluted in 3% BSA. Cells were then washed three times with PBS+ and incubated for 1 h at room temperature with the following secondary antibodies: donkey antibody against rabbit IgG coupled to Alexa Fluor 488 (Cat. A21206, 1/400) or to Alexa Fluor 555 (Cat. A32794, 1/400); donkey antibody against mouse coupled to Alexa Fluor 488 (Cat. A21202, 1/400) or Alexa Fluor 555 (Cat. A31570, 1/400); donkey antibody against sheep coupled to Alexa Fluor 488 (Cat. A11015, 1/400), and donkey antibody against goat coupled to Alexa Fluor 555 (Cat. A21432, 1/400). DAPI was utilized to stain the nuclei. Cells were washed three times, then coverslips were mounted onto slides with antifade reagent ProLong Glass (Invitrogen). Fluorescence images were acquired with a microscope Nikon A1 confocal inverted laser microscope at the University of Michigan Biomedical Research−Microscopy Core. Fluorescent co-localization index of cell-cell interfaces between HeLa cell co-cultures was performed using ImageJ software employing the Costes-related automatic threshold method followed by Pearson's R

correlation coefficient. Co-localization signal images were generated by calculating the fraction of interface signal between cells expressing different claudins that showed regions with a minimum of 500 contiguous nm with fluorescence intensity values greater than 100 (with an upper limit of 255) for both channels.

For STED microscopy, SKCO15 control and CLDN23 over-expressing cells were plated on #1.5H sterile coverslips (Cat. GG-12-1.5H-Pre), fixed in 4% paraformaldehyde in PBS for 30 min at 4 °C, and permeabilized with 0.5% Triton-X-100 before antibody incubations. Cells were incubated overnight at 4 °C with primary antibodies diluted in 3% BSA. Cells were then washed three times with PBS+ and incubated for 2 h at room temperature with the following secondary antibodies: goat antibody against rabbit IgG coupled to Abberior LLC STAR ORANGE (Cat. NC1933866, 1/100); goat antibody against mouse coupled to Abberior LLC STAR RED (Cat. NC1933868, 1/100). Images were taken in a Leica SP8 Confocal Microscope and analyzed by Huygens deconvolution software (Scientific Volume Imaging).

## Immunoblot
Immunoblotting for cell lines and isolated IECs was performed as described previously[99]. In short, cells were lysed in RIPA (20 mM Tris-Base, 150 mM NaCl, 2 mM EDTA, 2 mM EGTA, 1% sodium deoxycholate, 1% Triton X-100, 0.1% SDS, pH 7.4) containing protease and phosphatase inhibitor cocktails (Sigma-Aldrich), and protein concentration was determined using Pierce Protein BCA kit according to manufacturer's protocol. Samples were boiled at 100 °C for 10 min in NuPAGE LDS sample buffer (Life Technologies; Eugene, OR) with a final concentration of 100 mM DTT (Sigma-Aldrich) and 20 μg total protein was loaded onto polyacrylamide gels. After electrophoresis, the samples were transferred to a nitrocellulose membrane (Bio-Rad; Hercules, CA) and probed with primary antibodies diluted in 5% nonfat dry milk powder in Tris-buffered saline with 0.1% Tween-20. Membranes were then incubated with appropriate horseradish peroxidase (HRP)−conjugated secondary antibodies for 1 h at room temperature, followed by incubation with a chemiluminescence detection system (Clarity ECL Substrate). Finally, membranes were imaged by ChemiDoc imager (Bio-Rad).

## Red-RNAscope in situ hybridization
The localization of human and mouse CLDN23 mRNA was performed in either 6 or 8 μm sections (human and mouse, respectively) of formalin-fixed paraffin-embedded (FFPE) tissues. For human tissue staining, paraffin-embedded section slides of discarded resections from healthy patient screens were obtained with approval from the University of Michigan IRB. Slides were baked for 1 h at 60 °C, and deparaffinized. Followed by incubation with hydrogen peroxide (ACDbio, Cat. 322000) for 10 min at RT and subjected to antigen retrieval at 100 °C for 15 min. The tissue was digested by protease plus (Cat. 322330) for 15 min at 40 °C. CLDN23 probe for human (ACDbio, Cat. 563671) or mouse (ACDbio, Cat. 573311) were hybridized for 2 h at 40 °C followed by signal amplification. Chromogenic detection was performed accordingly with RNAscope® 2.5 HD Detection kit-RED (ACD, Cat. 322350). Slides were scanned with Aperio AT2.

## PCR
Total RNA was extracted from isolated IEC using the RNeasy Kit (QIAGEN) with on-column DNAse I treatment following the manufacturer's protocol. Total RNA (1 μg) was reverse transcribed into cDNA using iScript Reverse Transcription Supermix (Bio-Rad). Gene expression was analyzed by quantitative PCR (qPCR) using SYBR Green (Bio-Rad) with a Bio-Rad CTX Cycler measuring SYBR green incorporation for product detection. Reactions were performed in duplicate with at least five biological replicates. The primers sequence from humans was: GCCAGCAGCTTAATGGATTT and CGTCCAAAGGGT GGAATAGT for Cldn23. AGCATTGTGACAGCAGTTGG and GGGA GGAGATTGCACTGGAT for Cldn2, CAACCTGCA TGGACTGTGAAACC

and GTGGTCAAGTATTGGCGGTCAC for CLDN3, CTGCTTTGCTGC AACTGTCCAC and AGAGCGGGCAGCAGAATACTTG for CLDN4, CTTTGCTCTGCGGTTCTGA and GCTGGAGAAGGAGTTTCACTAC for CDX2, and TGC ACAGGAGCCAAGAGTGAA and CACATCACAGCTCC CCACCA for TBP. The relative expression was calculated by the $2^{\Delta\Delta Ct}$ method and normalized to the housekeeping gene TATA box-binding protein (TBP). The fold change was calculated by comparing the values to those obtained on control.

## Protein homology modeling and atomistic relaxation and coarse graining
The CLDN2, CLDN3, CLDN4, and CLDN23 structures were homology modeled using mClaudin-15 (4P79), hClaudin-4 (5B2G), hClaudin-9 (6OV2), mClaudin-19 (3×29), and mClaudin-3(3AKE) crystal structure in the YASARA molecular modeling software[100,101]. Templates were matched based on highest percent identity (>30%) with the sequence being modeled, as described previously[102–105]. Multiple sequence alignment (Supplementary Fig. 1) and identity matrix across modeled sequences and the respective crystal structures used are shown in Supplementary Fig. 7. For all in silico simulations and analyses, the C-terminal cytosolic tail portions of all claudins were truncated. The models were relaxed using MD simulations in all-atom resolution. The atomistic membrane-protein systems were built using the charmm-gui web server with 1-palmitoyl-2-oleoyl-glycero-3-phosphocholine (POPC) lipid bilayer, TIP3P water, and 0.15 M NaCl salt[106–108]. MD simulation was performed at 310.15 K and 1 bar pressure in NVT and NPT ensembles using V-rescale thermostat ($\tau_t = 1$ ps)[109] and Berendsen barostat ($\tau_p = 1$ ps) using GRO-MACS2018 molecular dynamics package[110]. The relaxed atomistic models were then coarse-grain (CG) mapped to a 4-1 mapping according to MARTINIv2.2 coarse-grain mapping scheme[111].

## Protein Association Energy Landscape (PANEL) method
*Cis* interaction landscapes for CLDN2, CLDN3, CLDN4 and CLDN23 in homomeric and heteromeric combinations were generated using the PANEL method following the implementation described previously[73]. Using in-house python scripts, PANEL outcomes were analyzed to obtain energy distributions and identify the stable dimer orientations of each *cis* dimer combination. The *cis* interaction energies lower than the 10th-percentile of the PANEL interaction energy distribution for each CLDN combination were extracted to represent the energy distribution of the most stable dimer orientations for each claudin combination studied in this work. These distributions were used to compare the *cis* interaction preference of each type of claudin combination over the others based on their interaction energies. The average energies of the top stable dimer configurations (which refers to the lowest energy values corresponding to lower than the 10th percentile value) were presented as bar plots with their error bars representing standard deviation. One-way ANOVA, followed by Tukey's posttest was used to perform hypothesis testing to compare CLDN dimer energy distributions, implemented using SciPy python package[112]. Each distribution comprised 12960 energy states, representing their respective top stable orientations.

## Pore structure modeling
Pore structures corresponding to each CLDN combination involving CLDN-23, CLDN-2, CLDN-3 and CLDN-4 were modeled using CLDN-15 channel forming *trans* interaction (tetramer) model as a template. The models were constructed in YASARA molecular modeling software using stable dimers obtained from PANEL analysis[100,101]. Two copies of stable dimers were placed symmetrically, such that they would interact head-on via their ECL regions, mimicking the tetrameric configuration of CLDN15 pore geometries. However, it was ensured to preserve the individual dimer geometries in order to retain the stable configurations unique to different CLDN partners. The symmetric placement of two dimers resulting into tetramer was further optimized using VINA

local docking implemented in YASARA[113]. Details regarding different modes of *trans* interactions: homotypic and heterotypic *trans* involving homomeric and heteromeric *cis* were adapted from the proposed experimental models. For further energy minimization and relaxation, the *trans* tetramer was coarse-grained according to the MARTINIv2.2 coarse-grain mapping scheme[111,114]. MD simulation was performed on the tetramer structures after introducing two sets of membranes aligned with the TM domains of each of the two *cis* dimers forming the tetramer, mimicking the adjacent cell membranes. The CG lipid membranes comprised of POPC lipids were introduced using insane.py script[115]. Solvent comprised of MARTINI water models, W and WF in 9:1 ratio and 0.15 M NaCl.

## Pore MD simulation

The membrane-embedded CLDNs were equilibrated with position restraints on the protein backbone in isochoric-isothermal (NVT) and isobaric-isothermal NPT ensembles for 75 ns. Production MD run was performed in the NPT ensemble for 2 microseconds with protein position restraints in the alpha helix backbone, while the extracellular domains were free of restraints to allow relaxation of the pore channel. The temperature during equilibration and production runs was maintained at 310 K using V-rescale thermostat ($\tau_t = 1\,ps$), and the pressure was maintained at 1 bar using Berendsen barostat during equilibration ($\tau_p = 1\,ps$) and using Parrinello–Rahman barostat during the production MD run ($\tau_p = 1\,ps$)[109,116].

## Pore diameter computation

The tetramer structures were extracted after the MD simulation and reverse mapped to atomistic resolution. Short minimizations were performed to relax the side chain orientations in reverse mapped structures. The resulting tetramers were imported to PyMol and the pore diameter was computed using the CAVER plugin[117,118].

## Pore lining residue evaluation

The final MD simulation frame of the tetrameric pore structures were post-processed using "gmx trjconv," a GROMACS utility function, to extract the claudins after removing the solvent and membrane environment for further analysis[119]. The tetrameric structures were converted to PDB file format and loaded into Pymol, where the channel was investigated using Caver plugin. After prediction of channels by Caver for each of the structures, the residues lining the pore channel were identified based on proximity of the residue sides chains to the channel with a 6 Å cut-off distance criterion. The residues identified with side chains close to the pore channel were rendered to be influencing the channel transport and the charged and polar residues were highlighted in sphere representation in Pymol. Further, the pore lining residues were segregated as positively charged (blue), negatively charged (red) and polar (green) as presented in Fig. 8 in the main section. The charged pore lining residue were collected to determine the net charge in each of the claudin tetramer combinations.

## *Trans* interaction energy measurements

*Trans* interaction energy was computed by grouping the individual *cis* dimers forming *trans* interaction as separate energy groups followed by a short MD for 5 ns with no position restraints. The resultant trajectory was used to measure the interaction energies using the "gmx energy" utility within GROMACS2018 MD package[119].

## Statistics

The statistical significance was measured by two-tailed Student's *t* test, one-way or two-way ANOVA with appropriate multiple comparisons posttest using Graphpad Prism software. A *p* value ≤ 0.05 was considered significant. Results are expressed as mean ± standard deviation (SD).

## Reporting summary

Further information on research design is available in the Nature Portfolio Reporting Summary linked to this article.

## Data availability

The source data, including original blots, immunofluorescence images, and quantification data, are provided as a Source data file. Source data are provided with this paper.

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

## Acknowledgements

The authors thank Dr. Roland Hilgarth for technical assistance in the generation of *Cldn23*^{f/f} and *Cldn23*^{ERΔIEC} mice, and HeLa cell clones expressing CLDN2 and CLDN23. We thank Dr. Mikio Furuse for advice on CLDN23 antibody generation and Dylan Fink for help with HeLa cell culture. We also thank the Transgenic and Gene Targeting Core at Emory University, Microscopy and Image Analysis Laboratory and the Unit for Laboratory Animal Medicine (ULAM) at the University of Michigan Medical School. This work was supported by the following grants: Crohn's and Colitis Foundation Research Fellowship no. 623536 (A.R.-S.), Training in Basic and Translational Digestive Sciences T32 DK094775 (K.M.L.-S.), National Science Foundation CAREER CBET-1453312 (S.N.), National Institutes of Health F30 DK132817 (K.M.L.-S.), R01 AA025854 and R01 HL158979 (M.K.), R01 DK61739 and DK72564 (C.A.P.), and R01 DK059888 and DK055679 (A.N.), RO1 129214 (A.N. and C.A.P.) and University of Michigan Center for Gastrointestinal Research (UMCGR), (NIDDK 5P30DK034933).

## Author contributions

A.R.-S., K.M.L.-S. and N.R. designed, performed experiments, analyzed, and interpreted data; A.R.-S. and K.M.L.-S. also drafted and edited the manuscript. J.B. and A.-C.L. helped with data analysis and interpretation, drafted, and edited the manuscript. V.G.-H. helped with design and generation of CLDN23 antibody and knockout mice, performed experiments and data analysis; A.N. supervised the study, designed experiments, provided resources, and helped with manuscript organization and editing. M.K., C.A.P. and S.N. helped with design of experiments and manuscript editing.

## Competing interests

The authors declare no competing interests.
