## [Peer Review File · Nature Communications]

REVIEWER COMMENTS

Reviewer #1 (Remarks to the Author):

Comments:

In this manuscript, the authors investigate the enrichment of an atypical claudin protein, claudin 23 (CLDN23), in luminal intestinal epithelial cells (IEC). In particular, they associated the presence of this protein to a reinforcement of the epithelial barriers. Moreover the authors discovered that loss of CLDN23 enhances the paracellular permeability. [LSEP]

In addition to the aforementioned findings, a detailed description of the heterogeneous interactions between CLDN23 and other two claudin homologs (claudin 4, CLDN4, and claudin 3, CLDN3) is presented.

The combination of results from in vivo and in vitro experiments is impressive. [LSEP] Structural modeling of the heteromeric complexes, based on the previously investigation of other CLDN complexes, is also shown. This computational investigation complemented the experimental findings. To the best of my knowledge this is the first computational attempt to describe heterogenous CLDN-based pore complexes.

Globally, this is a very interesting and nicely written work. Overall, the results of experiments and simulations are sound.

Once published, it will be a high quality example of a successful collaboration between experimental and computational scientists focused on the investigation of CLDN systems.

I recommend this article to be published in Nature Communications, but I ask that the following comments are addressed.

MAJOR COMMENTS:

1. I think that it should be important to add a multiple sequence alignment among the most representative CLDN proteins and including the atypical CLDN23. This, in order to further highlight the differences between the conventional members of the family and CLDN23. It might be included in the Supplementary Information file.

2. I do not agree with the univocal classification of CLDN-4 as a protein that prevents the passage of cations, but permissive to anions through the paracellular space (as stated in the INTRODUCTION). The structural functions of this protein are quite unclear and there is a dissensus about its classification. I think that this important detail should be included.

3. IN-SILICO MODELING

Regarding the introduction of the β -barrel structure which forms the paracellular pores (Ref. 8, 39 and 45),

a more detailed description of this model for non-expert readers should be highly appreciated.

Overall, the publication and validation of this model is the result of an intensive work of various and independent research groups that should be mentioned in the bibliography (if not yet in the submitted version).

Before the publication of Ref. 45, the following two works focused on the validation of the same model were published

* Zhao, J.; Krystofiak, E. S.; Ballesteros, A.; Cui, R.; Van Itallie, C. M.; Anderson, J. M.; Fenollar-Ferrer, C.; Kachar, B. Multiple Claudin-Claudin Cis Interfaces Are Required for Tight Junction Strand Formation and Inherent Flexibility. *Commun. Biol.* 2018, 1, 50.

* Alberini, G. ; Benfenati, F.; Maragliano, L. A refined model of claudin-15 tight junction paracellular architecture by molecular dynamics simulations. *PLoS One* 12, e0184190 (2017). (Already included as reference 53)

In addition, the following improvements and refinements of the model should be included and briefly mentioned.

* Irudayanathan, F.J. et al. Self-Assembly Simulations of Classic Claudins-Insights into the Pore Structure, Selectivity, and Higher Order Complexes. *J Phys Chem B* 122, 7463- 7474 (2018). (Already included as reference 43)

* Irudayanathan, F.J., Wang, N., Wang, X. & Nangia, S. Architecture of the paracellular channels formed by claudins of the blood-brain barrier tight junctions. *Ann N Y Acad Sci* 1405, 131-146 (2017). (Already included as reference 41)

* Fuladi, S.; McGuinness, S.; Khalili-Araghi, F. Role of TM3 in Claudin-15 Strand Flexibility: A Molecular Dynamics Study. *Front. Mol. Biosci.* 2022, 9 .

* Fuladi, S.; McGuinness, S.; Shen, L.; Weber, C. R.; Khalili-Araghi, F. Molecular Mechanism of Claudin-15 Strand Flexibility: A Computational Study. *J. Gen. Physiol.* 2022, 154 .

* Alberini, G.; Benfenati, F.; Maragliano, L. Molecular Dynamics Simulations of Ion Selectivity in a Claudin-15 Paracellular Channel. *J. Phys. Chem. B* 2018, 122, 10783.

* Berselli, A.; Alberini, G.; Benfenati, F.; Maragliano, L. Computational Assessment of Different Structural Models for Claudin-5 Complexes in Blood–Brain Barrier Tight Junctions. *ACS Chem. Neurosci.* 2022, 13, 2140–2153.

* Berselli, A.; Alberini, G.; Benfenati, F.; Maragliano, L. Computational Study of Ion Permeation through Claudin-4 Paracellular Channels. *Ann. N.Y. Acad. Sci.* 2022, 1516, 162–174.

* Irudayanathan, F.J. & Nangia, S. Paracellular Gatekeeping: What Does It Take for an Ion to Pass Through a Tight Junction Pore? *Langmuir* 36, 6757-6764 (2020). (Already included as reference 54)

4. In the following references

* Irudayanathan, F.J., Wang, N., Wang, X. & Nangia, S. Architecture of the paracellular channels formed by claudins of the blood-brain barrier tight junctions. *Ann N Y Acad Sci* 1405, 131-146 (2017). (Already included as reference 41)

* Irudayanathan, F.J. et al. Self-Assembly Simulations of Classic Claudins-Insights into the Pore Structure, Selectivity, and Higher Order Complexes. *J Phys Chem B* 122, 7463- 7474 (2018). (Already included as reference 43)

* Rajagopal N. et al. Predicting Selectivity of Paracellular Pores for Biomimetic Applications. Mol. Syst. Des. Eng., 2020, 5, 686

the computational authors of this manuscript described also another promising configuration (named Pore II), in order to study different CLDN-based paracellular cavities.

Why this second model was not considered for the description of the interactions between CLDN23 and the other CLDN members?

5. The idea to model a paracellular channel which globally preserves the D2 symmetry, but formed by different subunits is very fascinating. Are there any examples of other conventional channels formed by structurally different subunits?

I think that this observation and few examples (if any) should be important to further corroborate the structural modeling.

METHODS

The section related to the computational part lacks different references.

1. Protein homology modeling and atomistic relaxation and coarse graining. In Order to evaluate the quality of the homology modeling, I was wondering if it should be possible to have the sequence identity between the amino-acid composition of the template (murine CLDN15, PDB 4P79) and the target (CLDN23).

2. I am little confused about the use of two methods to assemble the paracellular tetramers, i.e.

iATTRACT - with the results in Figure 3

and

the assembly of different pores starting from the CLDN15 pore template, after PANEL (as illustrated in the Methods).

Why two methods?

3. I think that the references of the following methods/software should be addressed in the Bibliography.

- Reference(s) of YASARA

- I-ATTRACT (not discussed in the methods)

- the charmm-gui web server reference(s)

-----^[1]_{SEP}

S. Jo, T. Kim, V.G. Iyer, and W. Im (2008)

CHARMM-GUI: A Web-based Graphical User Interface for CHARMM. *J. Comput. Chem.* 29:1859-1865

J. Lee, X. Cheng, J.M. Swails, M.S. Yeom, P.K. Eastman, J.A. Lemkul, S. Wei, J. Buckner, J.C. Jeong, Y. Qi, S. Jo, V.S. Pande, D.A. Case, C.L. Brooks III, A.D. MacKerell Jr, J.B. Klauda, and W. Im (2016) CHARMM-GUI Input Generator for NAMD, GROMACS, AMBER, OpenMM, and CHARMM/OpenMM Simulations using the CHARMM36 Additive Force Field. *J. Chem. Theory Comput.* 12:405-413

And for the charmm-gui membrane builder, if applicable.

^[1]_{SEP}E.L. Wu, X. Cheng, S. Jo, H. Rui, K.C. Song, E.M. Dávila-Contreras, Y. Qi, J. Lee, V. Monje-Galvan, R.M. Venable, J.B. Klauda, and W. Im (2014)^[1]_{SEP}CHARMM-GUI Membrane Builder Toward Realistic Biological Membrane Simulations. *J. Comput. Chem.* 35:1997-2004

MINOR COMMENTS // MINOR CORRECTIONS

METHODS

Protein homology modeling and atomistic relaxation and coarse graining. structures were modeled *using* homology modeling *using* mCLDN-15 crystal structure (4P79).

Reviewer #2 (Remarks to the Author):

This is an ambitious and comprehensive study on a so far not well-characterized tight junction (TJ) protein, claudin-23 (CLDN23) and therefore potentially of very high interest. It is shown that CLDN23 is necessary to maintain the TJ integrity in colon epithelium as well as in organoid cultures. It is concluded that CLDN23 interacts heterotypically with either CLDN3 or CLDN4. Computer-based modeling as well as proximity ligation assay claim heterotypic associations CLDN23/CLDN3 and CLDN23/CLDN4.

The computational analysis has the potential to provide valuable information about TJ channel and barrier architecture. However, the functional studies and the molecular data and interpretations are too speculative and not sufficiently verified to justify the conclusions drawn.

Major issues

(1.) The functional studies and interpretations do not justify the assignment of observed permeability changes to the respective molecular pore changes. It would be necessary to add profound experiments on details of ion selectivity and permeability properties.

Fig. 2 E,F, H, I: If one compares the FD4 data with the TER changes, surprisingly both are altered to roughly the same ratios, namely by a factor between two and three.

If changes in pore dimensions and/or selectivity (as shown in Fig. 5) would be causative for the observed TER changes, the FD4 permeability should have remained unaltered. Of course, because of its size FD4 should not be able to travel through the pores of claudin-based paracellular channels. Therefore, the measured TER and FD4 changes cannot be functionally related to molecular pore changes due to CLDN23.

This would be expected only for sole changes in the so-called unrestricted pathway, which is based on (i) opening of large gaps in TJ strand continuity or (ii) by an increase of overall permeability of the tricellular TJ pathway. Regarding (i), to judge on CLDN23-related formation of gaps within the strand meshwork, morphometry by freeze-fracture electron microscopy would be the gold-standard method. Regarding (ii), to judge on CLDN23-related changes of the tricellular TJ, expression of their typical proteins should be measured as well as permeabilities for macromolecules of different sizes.

Technically, for FD4 results it is assumed that FD4 was dialyzed before experimentation in order to remove small fragments – if not, FD4 results may be erroneously high.

(2.) Positively thinking, to demonstrate that changes in the pores of claudin-based paracellular channels are causative for changes in ion permeability, TER alone is a too rough parameter. Because all known claudin-based channels are either specific for small cations or for small anions (and in two cases for water), at least the charge selectivity should be determined. For characterizing the channel properties in more detail, Eisenman sequences should be determined.

(3.) Cell cultures: It is unclear whether the cell cultures were just transient overexpressions and knockdowns or stable ones. In the methods part, something is said about selection with the associated antibiotics, but only for three days and then come the tests. Even assuming that ideally only the transfected cells are left, it would be good to show the uniform KD or the uniform overexpression in stains here as well and to look at potential changes in other TJ proteins, especially if an effect on CLDN3 and CLDN4 is assumed in other experiments.

If these are stable clones, then more than one clone should be shown, or if they are all that similar, mention that this is pooled data from different clones.

(4.) Animals: In the mice, too, control analyses are lacking which show that the KO of CLDN23 does not also change other TJ proteins, possibly even only regarding their localization (again, CLDN3 and CLDN4 as important candidates).

(5.) Fig. 3 C: How do the authors exclude that the observed differences in co-localization index are not mainly due to differences in the expression level and/or abundance in the plasma membrane for the different claudins? In addition, the signals in the images seem to be largely saturated resp. overexposed, raising the question about the accuracy of the quantification.

(6.) Fig. 3 D,E and text: The docking and MD approach used to explore whether CLDN23 preferentially interacts in trans with either CLDN3, CLDN4, or itself is comprehensible and based on methodology used already previously and partly even established by the authors. However, it is highly questionable, if the methods and data are reliable enough to justify the author's conclusions. The analysis was restricted to tetramers. As a consequence, interaction with neighboring claudins present in native TJs are lacking completely. To compensate the potential destabilization of the tetramers by this lack, the proteins were largely fixed at their positions using constraints of the non-ECL protein backbone. This limits the evaluation of the stability of the tetrameric structure and also restricts the protein flexibility. For example, it is questionable if the claudins studied here (claudin-3 as a barrier-forming claudin, claudin-4 with questionable pore properties, claudin-23 with unclear pore properties) have exactly the same positioning of the transmembrane helices as the clearly pore-forming claudin-15 that was used as a template.

Then again, flexibility of the ECLs is not restricted by the presence of neighboring claudins, as it is most likely the case in native strands. Indeed, the neighboring claudin molecules were indicated to critically influence the conformation of the "pore-forming" tetramer used here (Alberini et al. 2017; Samantha et al. 2018; Zhao et al. 2018; Hempel et al. 2020; Fuladi et al. 2022).

Trans interaction energy: For comparison, values for claudins, clearly shown to form homomeric/homotypic interactions, e.g. claudin-3/claudin-3 or claudin-15/claudin-15 are missing. Also negative controls are not provided.

(7.) "Furthermore, the Coulombic interaction energy between CLDN-23 and -2 showed a positive value (5.8 kJ/mol +/- 1.9) among the compared combinations, denoting repulsion between the interacting entities" "CLDN-23 and -2 trans interaction structure shows proximal placement of negatively charged residues GLU73 and ASP143 from CLDN-23 with ASP154 from CLDN-2."

Similar to claudin-2, claudin-3 has a negatively charged residue at the position corresponding to ASP154 in claudin-2. Nevertheless, a much lower Coulombic interaction energy was obtained for claudin-3/-23. How is this explained? Is there a trans-interaction between the region corresponding to ASP154 in claudin-2 and ASP-143 in claudin-23 for claudin-2, or -3 or -4 homo-tetramers? If not, why is an interaction between -2/-23 in this region expected? If yes, does this not block the pore entrance?

That claudin-4/claudin-23 shows the lowest trans interaction energy but the 2nd highest Coulombic interaction energy raises questions, too. How is this explained, in comparison to the other claudin pairs?

(8.) Fig. 4 and text: The depicted cis-models correspond just to one out of many small spots in the interaction energy landscapes and these spots do not correspond to the lowest energy. Thus, what is the relevance of the PANEL analysis data? How does this help to differentiate between different dimer conformations of similarly low energies? How can it be used to compare the stability or probability of "pore-forming" cis-dimers for different claudin pairs, if the difference in their energy values is not clearly higher than the energy difference between the many potential dimers reflected by interaction energy landscapes?

In addition, the energy of all the "non pore-forming" cis-dimers is much lower than the energy for the "pore-forming" cis-dimers. The relevance of these values and of the comparison is unclear. How do the non pore-forming cis-dimers look like, they are not shown? Is it suggested that claudin-3 homo-dimers and claudin-4 homo-dimers form "non pore-forming" dimers? Then why "pore-forming" dimers were used for the analysis of trans-interaction (Fig. 3) and pore-diameters (Fig. 5)?

(9.) Fig. 5 and text: Given the comments on Figure 3 and 4, the reliability of the pore models shown here is also questionable. The data suggest that claudin-4 as well as claudin-3 homo-tetramers form ion permeable pores. However, for claudin-4, the data about pore formation is inconsistent, and for claudin-

3 there are several study strongly arguing against channel formation by claudin-3. This is also very likely to be the case for homo-polymeric claudin-3 TJ-like strands (Gonschior et al. 2022, Nat Commun).

Other main points

(10.) Throughout the manuscript, "channel" and "pore" are used arbitrarily. By definition, a channel is the entire molecular structure comprising pore, size-limiting site, charge-selective site, and regulatory site. That "channel" and "pore" are different terms is convincingly illustrated by the existence of two-pore channels. The two terms should be used in a specific way whenever the one or the other is meant.

(11.) Within this, I don't like one of the explanations: "... CLDN2 is classified as a pore-forming or "leaky" CLDN, while CLDN3 and CLDN4 are barrier-forming or "tight" CLDNs" (line 79, also 316). This is misleading for a novice reader for two reasons: (i) The terms "leaky" and "tight" are classically assigned to properties of an entire epithelium exerting higher para- than transcellular ion conductance (leaky), or the opposite (tight). (ii) "Leaky claudins" sounds as if these are just non-perfectly sealing claudins instead of selective channels. Just "channel-forming claudins" and "barrier-forming claudins" would be most lucid.

(12.) Throughout the text, "permeability" or "paracellular permeability" was used without specification. However, the scientific term "permeability" makes sense only if referring to a substance or a group of substances. Here, "permeability" refers to only one macromolecular molecular size and thus should be named "FD4 permeability" or similar.

(13.) Line 661 "Paracellular permeability was done ...": What was actually done is a measurement of FD4 concentration increase within 3 h and the result is expressed as % change versus control (NS). For comparison with other studies and molecules, real FD4 permeabilities should be calculated and given in absolute numbers (cm/s).

(14.) At least some of the confocal microscopical images should include the Z-axis to ensure proper (co-)localization of the claudins within the apical TJ region.

Minor

(15.) Introduction: For introducing what is known about claudin-23, the few existing papers on that protein should be briefly mentioned, e.g. Katoh & Katoh 2003, Int J Mol Med; Wang et al. 2010, Mol Med Rep; Maryan et al. 2015, Mol Med Rep; Lu-Y et al. 2017, PlosOne.

(16.) Line 64: "The TJ is composed of ..." change to "The TJ between two cells is composed of ..." because the ensuing list of proteins describes the bicellular TJ only.

(17.) Line 160: That IECs form a barrier is no indicator for CLDN23 being a sealing protein. Also epithelia containing charge-selective channel-forming claudins are barrier-forming for all other solutes.

(18.) Throughout the text and Figures, CLDNs of numbers ## are written arbitrarily as "CLDN##" or as "CLDN-##", sometimes even within one sentence. In contrast to the full name "claudin-##", commonly no hyphens are used for "CLDN##".

(19.) Add hyphens, e.g. line 33: stem cell-derived, 174: CLDN23-depleted, 183: CLDN23-containing, 185 shRNA-mediated. Please check for all occasions throughout the text.

Reviewer #3 (Remarks to the Author):

The manuscript titled "Claudin-23 Regulates Epithelial Barrier Function by Altering Paracellular Pore Geometry" submitted for publication by Raya-Sandino et al describes the characterization of a novel claudin, CLDN23, whose function had been previously unknown. The manuscript is extremely well-written and the experiments are well designed, executed, and validated with proper controls and statistics. The conclusions are appropriate and based on the data presented. This work utilizes genetically modified mice, cell lines, and organoid cultures to demonstrate that CLDN23 plays a role in barrier function via its interactions with other claudin protein family members. Furthermore, they authors demonstrate that CLDN23 controls paracellular pore size which in turn affects permeability and thus is a substantial increase in knowledge surrounding the function of CLDN23. In particular, the finding that the neutrally charged amino acid in CLDN23 plays a role in its epithelial barrier function is interesting and supports previously published data on other claudins that contain similar residues. However, many claudins are spatially expressed and play a role in barrier function and although this was not known for CLDN23, the effect on paracellular pore size seems to be only an incremental increase in understanding the role claudins play in intestinal physiology. Others seem to have previously reported that claudins can effect pore size (Hempel et al 2022, Alberini et al 2017). As Nature Communications has a wide and vast readership of individuals across the scientific community, a more significant advancement seems warranted.

Comments:

- There was no prediction of a novel or unique role for CLDN23 and other hypotheses could have been explored such as a role in extracellular matrix interactions, proliferation, cell signaling, and differentiation.
- Of particular interest would have been characterization of the mechanisms that control the spatial expression of CLND23.
- Relating the loss of CLND23 expression to specific diseases would increase the relevance of the findings. There is no discussion of any phenotypic findings in the genetically modified mouse model and so are they assumed to be normal?
- The discussion mainly reiterates the results without really discussing how the data significantly advances the field. What will be possible now that this data is known?
- With the widespread use of human colonic organoids in the GI field, the use of cell lines in the paper seems outdated and not state of the art.
- Human colonic organoids can be genetically modified using CRISPR-CAS9 approaches and a series of experiments in this area would have greatly increased the impact of this paper.
- Some of the use of SEM seems inappropriate based on the samples size (less than 10).

Reviewer 1

Comments to the authors: In this manuscript, the authors investigate the enrichment of an atypical claudin protein, claudin 23 (CLDN23), in luminal intestinal epithelial cells (IEC)... ..The combination of results from in vivo and in vitro experiments is impressive. Structural modeling of the heteromeric complexes, based on the previously investigation of other CLDN complexes, is also shown. This computational investigation complemented the experimental findings. To the best of my knowledge this is the first computational attempt to describe heterogenous CLDN-based pore complexes.... Globally, this is a very interesting and nicely written work. Overall, the results of experiments and simulations are sound. Once published, it will be a high-quality example of a successful collaboration between experimental and computational scientists focused on the investigation of CLDN systems.

Major comments:

1. *I think that it should be important to add a multiple sequence alignment among the most representative CLDN proteins and including the atypical CLDN23. This, in order to further highlight the differences between the conventional members of the family and CLDN23. It might be included in the Supplementary Information file.*

Response: Based on this comment multiple sequence alignments of CLDN23 with a subset of classic and non-classic claudins has been added to the **Extended Data Fig. 1**.

2. *I do not agree with the univocal classification of CLDN-4 as a protein that prevents the passage of cations, but permissive to anions through the paracellular space (as stated in the INTRODUCTION). The structural functions of this protein are quite unclear and there is a dissensus about its classification. I think that this important detail should be included.*

Response: We thank the reviewer for the insightful comment. While claudin-4 has been considered predominantly a barrier-forming claudin, there is a dissensus regarding its paracellular ion flux selectivity which is context dependent. We have clarified this point in the Introduction (page 3) and the Discussion sections (page 17), and have included the following citations:

- Hou J, Gomes AS, Paul DL, Goodenough DA. Study of claudin function by RNA interference. *J Biol Chem* 281: 36117–36123, 2006
- Hou J, Renigunta A, Yang J, Waldegger S. Claudin-4 forms paracellular chloride channel in the kidney and requires claudin-8 for tight junction localization. *Proc Natl Acad Sci USA* 107: 18010–18015, 2010
- Van Itallie C, Rahner C, Anderson JM. Regulated expression of claudin-4 decreases paracellular conductance through a selective decrease in sodium permeability. *J Clin Invest* 107: 1319–1327, 2001
- Shashikanth N, France M, Xiao R, Haest X, Rizzo HE, Yeste J, Reiner J, Turner JR. Tight junction channel regulation by interclaudin interference. *Nat Commun* 13(1):3780, 2022

3. Regarding the introduction of the β -barrel structure which forms the paracellular pores (Ref. 8, 39 and 45), a more detailed description of this model for non-expert readers should be highly appreciated.

Response: We appreciate this comment by the reviewer. We have provided a more detailed description of this model in the revised manuscript on page 13.

3.1. Overall, the publication and validation of this model is the result of an intensive work of various and independent research groups that should be mentioned in the bibliography (if not yet in the submitted version).

Before the publication of Ref. 45, the following two works focused on the validation of the same model were published:

- Zhao, J.; Krystofiak, E. S.; Ballesteros, A.; Cui, R.; Van Itallie, C. M.; Anderson, J. M.; Fenollar-Ferrer, C.; Kachar, B. Multiple Claudin-Claudin Cis Interfaces Are Required for Tight Junction Strand Formation and Inherent Flexibility. *Commun. Biol.* 2018, 1, 50.
- Alberini, G. ; Benfenati, F.; Maragliano, L. A refined model of claudin-15 tight junction paracellular architecture by molecular dynamics simulations. *PLoS One* 12, e0184190 (2017). (Already included as reference 75)

Response: We agree with the reviewer, and we have added the Alberini G et. al reference to the revised version of the manuscript on page 13. Respectfully, the reference Zhao et al., 2018 was not included as it refers to the bending of the TJ strands, an aspect that has not been investigated in the revised manuscript.

3.2. In addition, the following improvements and refinements of the model should be included and briefly mentioned.

- Irudayanathan, F.J. et al. Self-Assembly Simulations of Classic Claudins-Insights into the Pore Structure, Selectivity, and Higher Order Complexes. *J Phys Chem B* 122, 7463- 7474 (2018). (Already included as reference 63)
- Irudayanathan, F.J., Wang, N., Wang, X. & Nangia, S. Architecture of the paracellular channels formed by claudins of the blood-brain barrier tight junctions. *Ann N Y Acad Sci* 1405, 131-146 (2017). (Already included as reference 61)
- Fuladi, S.; McGuinness, S.; Khalili-Araghi, F. Role of TM3 in Claudin-15 Strand Flex- ibility: A Molecular Dynamics Study. *Front. Mol. Biosci.* 2022, 9.
- Fuladi, S.; McGuinness, S.; Shen, L.; Weber, C. R.; Khalili-Araghi, F. Molecular Mechanism of Claudin-15 Strand Flexibility: A Computational Study. *J. Gen. Physiol.* 2022, 154 .
- Alberini, G.; Benfenati, F.; Maragliano, L. Molecular Dynamics Simulations of Ion Selectivity in a Claudin-15 Paracellular Channel. *J. Phys. Chem. B* 2018, 122, 10783.
- Berselli, A.; Alberini, G.; Benfenati, F.; Maragliano, L. Computational Assessment of Different Structural Models for Claudin-5 Complexes in Blood– Brain Barrier Tight Junctions. *ACS Chem. Neurosci.* 2022, 13, 2140–2153.

- Berselli, A.; Alberini, G.; Benfenati, F.; Maragliano, L. *Computational Study of Ion Per- meation through Claudin-4 Paracellular Channels*. *Ann. N.Y. Acad. Sci.* 2022, 1516, 162–174.
- Irudayanathan, F.J. & Nangia, S. *Paracellular Gatekeeping: What Does It Take for an Ion to Pass Through a Tight Junction Pore?* *Langmuir* 36, 6757-6764 (2020). (Already included as reference 54)

Response: Some of the above references were cited in our manuscript and the remaining references have been added to the revised paper.

4. In the following references

- Irudayanathan, F.J., Wang, N., Wang, X. & Nangia, S. *Architecture of the paracellular channels formed by claudins of the blood-brain barrier tight junctions*. *Ann N Y Acad Sci* 1405, 131-146 (2017). (Already included as reference 41)
- Irudayanathan, F.J. et al. *Self-Assembly Simulations of Classic Claudins-Insights into the Pore Structure, Selectivity, and Higher Order Complexes*. *J Phys Chem B* 122, 7463-7474 (2018). (Already included as reference 43)
- Rajagopal N. et al. *Predicting Selectivity of Paracellular Pores for Biomimetic Applications*. *Mol. Syst. Des. Eng.*, 2020, 5, 686

the computational authors of this manuscript described also **another promising configuration** (named Pore II), in order to study different CLDN-based paracellular cavities.

Why was this second model not considered for the description of the interactions between CLDN23 and the other CLDN members?

Response: The second or pore II model has been generated from dimer type B as described in a previous publication (Irudayanathan et al., 2018). The dimer B *cis* configuration corresponds to the $180^\circ \times 180^\circ$ position on the PANEL landscape. As the PANEL landscapes for heteromeric *cis* interactions between CLDN23 and CLDN2, CLDN3, or CLDN4 (see **Fig. 1** below) did not have higher frequency in the $180^\circ \times 180^\circ$ region (**white arrow**), the probability of pore II formation was considered low and was not explored further in the manuscript.

Figure 1. PANEL frequency plots for CLDN23 with CLDN2 (left), CLDN3 (middle), and CLDN4(right). Dimer B (white arrow) and dimer D (red arrow) regions are indicated on the plots.

7. *The idea to model a paracellular channel which globally preserves the D2 symmetry, but formed by different subunits is very fascinating. Are there any examples of other conventional channels formed by structurally different subunits? I think that this observation and few examples (if any) should be important to further corroborate the structural modeling.*

Response: We thank the reviewer for this comment. To expand on this idea of conventional channels formed by structurally different subunits, we have included new information on gap junctions and epithelial sodium channels (ENaCs). Gap junctions are formed by homomeric and heteromeric interactions between different connexin family members to form a hemichannel (connexon). Two hemichannels in adjacent cells have been observed to create functional heterotypic gap junctions. For example, ENaC channels are responsible for transcellular sodium reabsorption by kidney tubule epithelial cells. Canonical ENaCs are heterotrimers formed by α , β , and γ subunits. This new information is now included in the introduction section of the revised manuscript with the following relevant citations on page 4:

- Koval, M. Pathways and control of connexin oligomerization. *Trends in cell biology*. 16(3):159-166. (2006).
- Canessa, C.M. Schild, L., Buell, G., Thorens, B., Gautschi, I., Horisberger, J.D., et al. Amiloride-sensitive epithelial Na⁺ channel is made of three homologous subunits. *Nature*, 367: 463-467. (1994b).

8. *Protein homology modeling and atomistic relaxation and coarse graining. In Order to evaluate the quality of the homology modeling, I was wondering if it should be possible to have the sequence identity between the amino-acid composition of the template (murine CLDN15, PDB 4P79) and the target (CLDN23).*

Response: We used the following crystal structures for homology modeling: mClaudin-15 (4P79), hClaudin-4 (5B2G), hClaudin-9 (6OV2), mClaudin-19 (3X29), mClaudin-3(3AKE). We have added this information to the methods section of the revised the manuscript on page 34.

The crystal structures used as templates for homology modeling were ensured to satisfy >30% sequence identity requirement with the sequence to be modeled (**Extended data Fig. 7**). Claudin-23 has 33.49% identity with hClaudin-9 crystal structure. Claudin-2 – 39.5% identity with hClaudin-4 crystal structure. Claudin-3 – 91.3% identity with mClaudin-3 crystal structure. hClaudin-4 crystal structure was used for modeling Claudin-4. Moreover, the sequence identities were computed for the entire length of the sequence, although the C-terminal portions were not considered in this study. Excluding the C-terminal portion of the sequence may result in higher identities than those reported above.

The identity matrix has been added to the **Extended data Fig. 7**.

9. *I am little confused about the use of two methods to assemble the paracellular tetramers, i.e. iATTRACT - with the results in Figure 3 and the assembly of different pores starting from the CLDN15 pore template, after PANEL (as illustrated in the Methods). Why two methods?*

Response: We apologize for any confusion and have modified the Results and Methods section to clarify the method utilized to assemble the paracellular tetramers. The following text has been incorporated as described below:

Edited text in Results section (see page 11): “Computational modeling between *cis* homodimers of CLDN23 with *cis* homodimers of CLDN2, CLDN3, CLDN4, and CLDN23 (**Fig. 5D**) was conducted in YASARA (*details in methods section*) and coarse-grained MD simulations were performed to evaluate trans interaction energy and determine preferable binding.”

Edited text in Methods section (see page 34): “Pore structures corresponding to each CLDN combination involving CLDN23, CLDN2, CLDN3 and CLDN4 were modeled using CLDN15 channel forming trans interaction (tetramer) model as a template. The models were constructed in YASARA molecular modeling software by aligning the structures of desired claudins with the template CLDN15 structure.”

10. *I think that the references of the following methods/software should be addressed in the Bibliography.*

- Reference(s) of YASARA
- I-ATTRACT (not discussed in the methods)
- the charmm-gui web server reference(s)

S. Jo, T. Kim, V.G. Iyer, and W. Im (2008)

CHARMM-GUI: A Web-based Graphical User Interface for CHARMM. J. Comput. Chem. 29:1859-1865

J. Lee, X. Cheng, J.M. Swails, M.S. Yeom, P.K. Eastman, J.A. Lemkul, S. Wei, J. Buckner, J.C. Jeong, Y. Qi, S. Jo, V.S. Pande, D.A. Case, C.L. Brooks III, A.D. MacKerell Jr, J.B. Klauda, and W. Im (2016) CHARMM-GUI Input Generator for NAMD, GROMACS, AMBER, OpenMM, and CHARMM/OpenMM Simulations using the CHARMM36 Additive Force Field. J. Chem. Theory Comput. 12:405-413

And for the charmm-gui membrane builder, if applicable.

E.L. Wu, X. Cheng, S. Jo, H. Rui, K.C. Song, E.M. Dávila-Contreras, Y. Qi, J. Lee, V. Monje-Galvan, R.M. Venable, J.B. Klauda, and W. Im (2014) CHARMM-GUI Membrane Builder Toward Realistic Biological Membrane Simulations. J. Comput. Chem. 35:1997-2004

Response: YASARA and CHARMM-GUI references have been added as suggested by the reviewer.

Minor comments // Minor corrections:

1. *Protein homology modeling and atomistic relaxation and coarse graining. structures were modeled *using* homology modeling *using* mCLDN-15 crystal structure (4P79).*

Response: This sentence has been reworded based on the reviewer’s comment.

Reviewer 2

Comments to the authors: This is an ambitious and comprehensive study on a so far not well-characterized tight junction (TJ) protein, claudin-23 (CLDN23) and therefore potentially of very

high interest. The computational analysis has the potential to provide valuable information about TJ channel and barrier architecture. However, the functional studies and the molecular data and interpretations are too speculative and not sufficiently verified to justify the conclusions drawn.

1. *The functional studies and interpretations do not justify the assignment of observed permeability changes to the respective molecular pore changes. It would be necessary to add profound experiments on details of ion selectivity and permeability properties. Fig. 2E, F, H, I: If one compares the FD4 data with the TER changes, surprisingly both are altered to roughly the same ratios, namely by a factor between two and three. If changes in pore dimensions and/or selectivity (as shown in Fig. 7) would be causative for the observed TER changes, the FD4 permeability should have remained unaltered. Of course, because of its size FD4 should not be able to travel through the pores of claudin-based paracellular channels. Therefore, the measured TER and FD4 changes cannot be functionally related to molecular pore changes due to CLDN23. This would be expected only for sole changes in the so-called unrestricted pathway, which is based on (i) opening of large gaps in TJ strand continuity or (ii) by an increase of overall permeability of the tricellular TJ pathway. Regarding (i), to judge on CLDN23-related formation of gaps within the strand meshwork, morphometry by freeze-fracture electron microscopy would be the gold-standard method. Regarding (ii), to judge on CLDN23-related changes of the tricellular TJ, expression of their typical proteins should be measured as well as permeabilities for macromolecules of different sizes.*

Response (please note that we have additional data and therefore the figure numbers have changed in the revised manuscript):

We thank the reviewer for the thorough feedback and recommendations. In line with these suggestions, we have performed additional comprehensive experiments to further characterize the influence of CLDN23 on epithelial barrier function. Because our studies focus on the role of CLDN23 in intact epithelial monolayers, we have added additional experiments to better characterize CLDN23 channel permeability to ions through the ‘pore pathway’ (**Fig. 8 A-D, & Extended data Fig. 11**) and to macromolecules (4kDa and 70kDa dextrans) via the ‘leak pathway’ (**Fig. 2C, F & I**). We comment on our findings below, and in the Results section of the manuscript on pages 8 and 13-14, and in the Discussion section on page 15. Since the monolayers were intact with a high TEER and no difference 70kDa dextran flux, we did not analyze the unrestricted pathway that refers to a third, tight junction-independent, pathway for particles to cross the epithelial barrier owing to epithelial damage (Nalle & Turner, 2015; Van Itallie & Anderson, 2011).

Additionally, we have added real-time TEER measurements in SKCO15 cells overexpressing CLDN23 vs. control SKCO15 cells, and in T84 cells with CLDN23 knockdown, using a CellZScope system. Overexpression of CLDN23 increased transepithelial resistance that remained above controls for 5 days (**Fig. 2B**). In contrast, T84 cells lacking CLDN23 expression exhibited decreased TEER compared to CLDN23 expressing control cells (**Fig. 2E**). These changes in TEER over time correlate with pore pathway regulation and suggest claudin based control of ion permeability as described in section (c) below.

a) In regards with reviewer's comments;

“It would be necessary to add profound experiments on details of ion selectivity and permeability properties”.....“If changes in pore dimensions and/or selectivity (as shown in Fig. 5) would be causative for the observed TER changes, the FD4 permeability should have remained unaltered. Of course, because of its size FD4 should not be able to travel through the pores of claudin-based paracellular channels. Therefore, the measured TER and FD4 changes cannot be functionally related to molecular pore changes due to CLDN2”. This would be expected only for sole changes in the so-called unrestricted pathway”

Regarding Reviewer's comment “This would be expected only for sole changes in the so-called unrestricted pathway, which is based on (i) opening of large gaps in TJ strand continuity or (ii) by an increase of overall permeability of the tricellular TJ pathway”.

Response: We agree with the reviewer and we have performed additional experiments to analyze paracellular ion movement. Comprehensive ion dilution potential studies were performed by employing two different electrophysiologic approaches (EVOM and Ussing chamber) (**Fig. 8 A-D and Extended data Fig. 11**). The data demonstrated that paracellular permeability to Na⁺, Li⁺, and Cl⁻ ions were markedly reduced when CLDN23 was expressed in IECs (**Fig. 8, A-D and Extended data Fig. 11**, pages 13, 14). Additionally, we examined the nonrestrictive pathway by investigating paracellular flux of 4kDa and 70kDa dextran in IECs with CLDN23 knockdown and CLDN23 overexpression (**Fig. 2, C & F**, page 8).

Our results demonstrate that modulation of CLDN23 influences paracellular ion movement and 4kDa dextran flux, consistent with a role for CLDN23 in sealing tight junctions via both a pore altering pathway (ions) and through changes in strand architecture (4kDa dextran). CLDN23 is not likely to play a significant role in regulating tricellular junctions. However, we feel that a detailed analysis of tricellular junctions is beyond the scope of this study.

b) *Technically, for FD4 results it is assumed that FD4 was dialyzed before experimentation in order to remove small fragments – if not, FD4 results may be erroneously high.*

Response: We thank the reviewer for this comment. We have connected with Sigma Aldrich, and we have been assured that the FD4 dextran used in this study has >98% purity. This detail has been added to the methods section of the revised manuscript.

2. Positively thinking, to demonstrate that changes in the pores of claudin-based paracellular channels are causative for changes in ion permeability, TER alone is a too rough parameter. Because all known claudin-based channels are either specific for small cations or for small anions (and in two cases for water), at least the charge selectivity should be determined. For characterizing the channel properties in more detail, Eisenman sequences should be determined.

Response: We thank the reviewer for this thoughtful comment, and we agree that it is important to show changes in ion selectivity across epithelial cells upon CLDN23 overexpression or knockdown and, to determine Eisenman sequences.

a) **Fig.8, A-D:** To analyze paracellular permeability to small ions (Na⁺, Li⁺, and Cl⁻) we have performed dilution potential assays in CLDN23 overexpressing and in *Cldn23*

knockdown intestinal epithelial cells. We observed that overexpression of CLDN23 decreases permeability to cations (Li^+ and Na^+) and the Cl^- anion.

- b) Regarding **Eisenman sequences**, our findings with Li^+ and Na^+ paracellular flux (**Fig. 8, A-D**) show that CLDN23 paracellular channels restrict the passage of Na^+ more than that of Li^+ . This fits either the Eisenman sequence XI ($\text{Li}^+ > \text{Na}^+ > \text{K}^+ > \text{Rb}^+ > \text{Cs}^+$) or the non-Eisenman sequence X ($\text{Li}^+ > \text{Na}^+ > \text{Rb}^+ > \text{K}^+ > \text{Cs}^+$). Although we have not evaluated the permeability to Rb^+ , K^+ , and Cs^+ , given the relative abundance of channels that conform to the Eisenman sequence XI versus the non-Eisenman sequence X in vivo (61.40% vs. 1.04%, respectively (Krauss et al., 2011)), it is likely that the CLDN23 channel follows sequence XI. This is now included in the Discussion of the revised manuscript on pages 19-20 of the revised manuscript.
- c) To further expand on the data in **Fig. 8**, we have performed additional computational analyses to identify charged amino acid residues that line the CLDN channel described in **Fig. 7**. These new findings have been correlated with the dilution potential results. These new data have been added to the results and discussion sections of the revised manuscript (**Fig. 8, E-H** and pages 14 and 19).

3. Cell cultures: *It is unclear whether the cell cultures were just transient overexpressions and knockdowns or stable ones. In the methods part, something is said about selection with the associated antibiotics, but only for three days and then come the tests. Even assuming that ideally only the transfected cells are left, it would be good to show the uniform KD or the uniform overexpression in stains here as well and to look at potential changes in other TJ proteins, especially if an effect on CLDN3 and CLDN4 is assumed in other experiments. If these are stable clones, then more than one clone should be shown, or if they are all that similar, mention that this is pooled data from different clones.*

Response: Based on this comment we have better clarified experimental procedures for viral transduction (page 28). Intestinal epithelial cells were transduced with a viral vector containing either CLDN23 shRNA (using two different shRNA sequences), non-silencing shRNA (scramble sequence), full length human CLDN23 construct or a control construct containing a myc tag. We achieved a very high transduction efficiency (>80%) and used puromycin to select for transduced cells. At least 3 different batches of transduced cells were used for experiments. Furthermore, we have included details on the generation of stable HeLa cell line clones and have cited our previous manuscript showing that HeLa clones have comparable levels of induced claudin expression at the plasma membrane (Daugherty et al., 2007). We would also like to thank the reviewer for the insightful experimental suggestions that have led to the addition of the following new figure:

- a) **Extended data Fig. 4:** To demonstrate uniform KD or uniform overexpression of CLDN23 we have performed immunofluorescence staining and confocal microscopy in T84, SKCO15 epithelial cells and primary murine colonoids. We have also included immunostaining for CLDN4 in these CLDN23 overexpressing or CLDN23 depleted IEC monolayers.

4. Animals: *In the mice, too, control analyses are lacking which show that the KO of CLDN23 does not also change other TJ proteins, possibly even only regarding their localization (again, CLDN3 and CLDN4 as important candidates).*

Response: We thank the reviewer for this suggestion, and we have added the following new data to the revised manuscript:

- a) **Fig. 3 A & B:** To determine effects of CLDN23 expression on CLDN3, CLDN4, and ZO1 we have performed western blots of colonic tissue and differentiated colonoid monolayers derived from *Cldn23^{fl/fl}* and *Cldn23^{ERΔIEC}* mice. As shown in revised **Fig. 3** CLDN23 knockout does not modify expression of these proteins. We now comment on this in the Results section on page 9.
- b) **Fig. 3C:** To further examine the effects of CLDN23 expression on localization of other CLDN proteins we performed immunofluorescence labeling and confocal microscopy in primary murine colonoid cocultures derived from *Cldn23^{fl/fl}* and *Cldn23^{ERΔIEC}* mice. We observed that CLDN3 and CLDN4 are predominantly localized in the TJ plasma membranes in cells expressing CLDN23. However, in CLDN23 knockdown cells, CLDN3 and CLDN4 lack this sharp TJ localization and are distributed in the lateral plasma membrane and beneath the TJ in vesicle-like structures. Similar findings were observed in model intestinal epithelial cells with CLDN23 knockdown (**Extended data Fig. 4B**). We have added a comment in the Results and Discussion sections highlighting this change in distribution of CLDN3 and CLDN4 in the presence of CLDN23 on page 9-10 and 16.

5. *Fig. 3C: How do the authors exclude that the observed differences in co-localization index are not mainly due to differences in the expression level and/or abundance in the plasma membrane for the different claudins? In addition, the signals in the images seem to be largely saturated resp. overexposed, raising the question about the accuracy of the quantification.*

Response: We agree with the reviewer that differences in the CLDN expression level in the plasma membrane can influence the results of the CLDN/CLDN co-localization. We have performed western blots of HeLa cells expressing the respective CLDNs. As shown in **Extended data Fig. 2** similar levels of CLDNs 2, 3 and 4 protein were observed in cells used for the experiment.

6. *Fig. 3 D, E and text: The docking and MD approach used to explore whether CLDN23 preferentially interacts in trans with either CLDN3, CLDN4, or itself is comprehensible and based on methodology used already previously and partly even established by the authors. However, it is highly questionable, if the methods and data are reliable enough to justify the author's conclusions. The analysis was restricted to tetramers. As a consequence, interaction with neighboring claudins present in native TJs are lacking completely. To compensate the potential destabilization of the tetramers by this lack, the proteins were largely fixed at their positions using constraints of the non-ECL protein backbone. This limits the evaluation of the stability of the tetrameric structure and also restricts the protein flexibility. For example, it is questionable if the claudins studied here (claudin-3 as a barrier-forming claudin, claudin-4 with questionable pore properties, claudin-23 with unclear pore properties) have exactly the same positioning of the transmembrane helices as the clearly pore-forming claudin-15 that was used as a template.*

Response:

We used CLDN15 primarily as an example to inform the reader about the dimer building block as a starting point to construct CLDN tetramers. The dimers in the present work were obtained directly from PANEL results and were used for tetramer construction without any changes in dimer geometry or influence from CLDN15, thereby preserving the preferred dimer orientation unique to different CLDN partners.

The tetramers were constructed by the YASARA molecular modeling software using stable dimers obtained from PANEL results (Krieger & Vriend, 2014; Land & Humble, 2018). Two copies of stable dimers were placed symmetrically, such that they interact head-on via the extracellular segments (ECS), mimicking the tetrameric configuration of CLDN15 pore geometries. However, we made sure to preserve individual dimer geometries such that stable configurations unique to different CLDN partners were retained. The symmetric placement of two dimers resulting in tetramer generation was further optimized using VINA local docking in YASARA (Trott & Olson, 2010) (page 34).

Then again, flexibility of the ECLs is not restricted by the presence of neighboring claudins, as it is most likely the case in native strands. Indeed, the neighboring claudin molecules were indicated to critically influence the conformation of the "pore-forming" tetramer used here (Alberini et al. 2017; Samantha et al. 2018; Zhao et al. 2018; Hempel et al. 2020; Fuladi et al. 2022).

Response:

Fuladi et al. (Fuladi et al., 2022) state “*claudins are assembled into interlocking tetrameric ion channels along the strand that slide with respect to each other as the strands curve over submicrometer-length scales.*” This work highlights strand dynamics and the stability of tetrameric pores.

Additionally, Alberini et al. (Alberini et al., 2017) state “*Our double-pore system comprises structural features that complement those of the single-pore and that are responsible for TJ strand formation.*”

Both these studies show that tetrameric claudins form a stable “building block” for tight junction strands. The goal of our manuscript is to establish how CLDN23/CLDN3 and CLDN23/CLDN4 interact. Therefore, we provided information of the tetrameric building blocks that were highlighted by Alberini et al., 2017 and Fuladi et al., 2022. We feel that further characterization of strand architecture and dynamics are beyond the scope of the present work.

Trans interaction energy: For comparison, values for claudins, clearly shown to form homomeric/homotypic interactions, e.g. claudin-3/claudin-3 or claudin-15/claudin-15 are missing. Also, negative controls are not provided.

Response:

The non-bonded (*trans*) interaction energies are shown below for all claudin pairs, and have been also included the revised manuscript in graphs (**Fig. 5D and Extended data Fig. 8, A & B**):

Interactions (kJ/mol)	CLDN 2-23	CLDN 3-23	CLDN 4-23	CLDN 23-23	CLDN 3-3	CLDN 4-4
-----------	-----------	-----------	------------	----------	----------

LJ	-997.1 ±5.8	-1282.8 ±57.8	-1684.5 ±38.9	-1024.8 ±24.3	-805.91 ±64.8	-1666.4 ±49.4
Coulombic	5.8 ±1.4	-16.4 ±2.5	-0.2 ±0.5	-12.5 ±3.7	2.3 ±2.9	13 ±1.6
Nonbonding	-991.3 ±28.8	-1298.6 ±60.3	-1684.7 ±39.4	-1037.3 ±28.0	-803.6 ±67.7	-1653.4 ±51.0

"Furthermore, the Coulombic interaction energy between CLDN23 and 2 showed a positive value (5.8 kJ/mol +/- 1.9) among the compared combinations, denoting repulsion between the interacting entities" "CLDN23 and 2 trans interaction structure shows proximal placement of negatively charged residues GLU73 and ASP143 from CLDN23 with ASP154 from CLDN2." Similar to claudin-2, claudin-3 has a negatively charged residue at the position corresponding to ASP154 in claudin-2. Nevertheless, a much lower Coulombic interaction energy was obtained for claudin-3/-23. How is this explained? Is there a trans-interaction between the region corresponding to ASP154 in claudin-2 and ASP-143 in claudin-23 for claudin-2, or -3 or -4 homo-tetramers? If not, why is an interaction between -2/-23 in this region expected? If yes, does this not block the pore entrance? That claudin-4/claudin-23 shows the lowest trans interaction energy but the 2nd highest Coulombic interaction energy raises questions, too. How is this explained, in comparison to the other claudin pairs?"

Response:

The non-bonded interaction energy that includes the Lennard Jones (LJ) and Coulombic interactions determines the stability of the system. Since the LJ energies are two orders of magnitude lower than the Coulombic interactions for all trans models, these interactions play a dominant role in determining stability. Thus, Coulombic interactions are negligible in determining *trans* stability. These explanations have been added to the results and discussion sections of the revised manuscript (page 18).

8. Fig. 4 and text: *The depicted cis-models correspond just to one out of many small spots in the interaction energy landscapes and these spots do not correspond to the lowest energy. Thus, what is the relevance of the PANEL analysis data? How does this help to differentiate between different dimer conformations of similarly low energies? How can it be used to compare the stability or probability of "pore-forming" cis-dimers for different claudin pairs, if the difference in their energy values is not clearly higher than the energy difference between the many potential dimers reflected by interaction energy landscapes?*

Response:

The PANEL output contains multiple outputs, including dimer energies, populations, and free energies. We studied all stable dimers in each panel plot and determined if the ECS domains would lead to a pore or a tighter barrier. A systematic investigation of all claudin landscapes was performed to determine the key dimers presented in this work.

In addition, the energy of all the "non-pore forming" cis-dimers is much lower than the energy for the "pore-forming" cis-dimers. The relevance of these values and of the compassion is unclear. How do the non-pore forming cis-dimers look like, they are not shown? Is it suggested that claudin-3 homo-dimers and claudin-4 homo-dimers form "non pore-forming" dimers? Then why "pore-forming" dimers were used for the analysis of trans-interaction (Fig. 5) and pore-diameters (Fig. 7)?

Response:

As the manuscript title suggests, our focus was to provide evidence of how CLDN23 regulates epithelial barrier integrity by altering pore geometry/architecture. We expect that claudins can form non-pore forming *trans* interfaces. However, to demonstrate the effect of claudins in regulating epithelial barrier function, we provided the structure and energies of stable pores and combinations where those pores failed to form (**Fig. 7**).

9. *Fig. 5 and text: Given the comments on Figure 3 and 4, the reliability of the pore models shown here is also questionable. The data suggest that claudin-4 as well as claudin-3 homotetramers form ion permeable pores. However, for claudin-4, the data about pore formation is inconsistent, and for claudin-3 there are several studies strongly arguing against channel formation by claudin-3. This is also very likely to be the case for homo-polymeric claudin-3 TJ-like strands (Gonschior et al. 2022, Nat Commun).*

Response:

Regarding CLDN3, a study reported by Milatz et al. (Milatz et al., 2010) demonstrated that this claudin decreases permeability of ions of either charge as well as uncharged solutes. However, this finding does not exclude the presence of a pore. Paracellular pores are formed through the *trans* interactions, their size and shape would be influenced by TM3 bending, possibly affecting the barrier and permeation properties (Nakamura et al., 2019). Thus, there could be a pore, but permeability would be limited by non-pore factors.

In the Gonschior paper (Gonschior et al., 2022), the authors show that CLDN3 forms strands on its own, CLDN4 requires another claudin (e.g. CLDN3) to form strands (the integration model, Figure 3b). Even though CLDN4 does not form strands on its own, there could still be CLDN4 homotypic channels within heterogeneous strands consistent with our models. Additionally, consistent with this hypothesis, a computational study by Berselli et al (Berselli et al., 2022) showed two models for CLDN4 pores. One model has a cavity permeable to chloride and repulsive to cations, the other model shows a complete (non-pore forming) barrier to the passage of all major physiological ions.

10. *Edit Throughout the manuscript, "channel" and "pore" are used arbitrarily. By definition, a channel is the entire molecular structure comprising pore, size-limiting site, charge-selective site, and regulatory site. That "channel" and "pore" are different terms is convincingly illustrated by the existence of two-pore channels. The two terms should be used in a specific way whenever the one or the other is meant.*

Response: We thank the reviewer for this comment. We agree and we now refer to the claudin tetramer as a claudin “channel” and we utilize the term “pore” to refer specifically to the channel opening.

11. *Edit Within this, I don't like one of the explanations: "... CLDN2 is classified as a pore-forming or "leaky" CLDN, while CLDN3 and CLDN4 are barrier-forming or "tight" CLDNs" (line 79, also 316). This is misleading for a novice reader for two reasons: (i) The terms "leaky"*

and "tight" are classically assigned to properties of an entire epithelium exerting higher paracellular ion conductance (leaky), or the opposite (tight). (ii) "Leaky claudins" sounds as if these are just non-perfectly sealing claudins instead of selective channels. Just "channel-forming claudins" and "barrier-forming claudins" would be most lucid.

Response: We agree with the reviewer, and we have now changed the text to reflect "channel-forming claudins" and "barrier-forming claudins".

12. Edit Throughout the text, "permeability" or "paracellular permeability" was used without specification. However, the scientific term "permeability" makes sense only if referring to a substance or a group of substances. Here, "permeability" refers to only one macromolecular molecular size and thus should be named "FD4 permeability" or similar.

Response: We have modified the text to reflect "FD70 permeability" or "TD4 permeability".

13. Edit and experiment we have. Line 661 "Paracellular permeability was done ...": What was actually done is a measurement of FD4 concentration increase within 3 h and the result is expressed as % change versus control (NS). For comparison with other studies and molecules, real FD4 permeabilities should be calculated and given in absolute numbers (cm/s).

Response: As recommended, we have added new data to **Fig. 2, C and F** that presents calculated FD70 and TD4 permeabilities over time as apparent permeability coefficient (P_{app} ; units are cm/s). Apparent permeability was calculated using the following equation: $P_{app} = (dQ/dt) / (A \cdot C_0)$ where dQ/dt is the steady state flux (mmol/second), A is the surface area of the filter (cm^2) and C_0 is the initial concentration in the donor chamber (mmol/ cm^3).

14. At least some of the confocal microscopical images should include the Z-axis to ensure proper (co) localization of the claudins within the apical TJ region.

Response: We have now included Z-axis images of polarized epithelial cells to demonstrate distribution of CLDNs in TJs and the lateral membrane (**Extended data Fig. 4**).

Edit. Minor

15. Introduction: For introducing what is known about claudin-23, the few existing papers on that protein should be briefly mentioned, e.g. Katoh & Katoh 2003, Int J Mol Med; Wang et al. 2010, Mol Med Rep; Maryan et al. 2015, Mol Med Rep; Lu-Y et al. 2017, PlosOne.

Response: We now include the suggested references in the revised manuscript. We comment on these existing papers in page 5 of the revised manuscript.

16. Line 64: "The TJ is composed of ..." change to "The TJ between two cells is composed of ..." because the ensuing list of proteins describes the bicellular TJ only.

Response: We now focus our introduction section on claudin proteins in TJs and we have removed reference to other proteins in the bicellular TJ.

17. Line 160: That IECs form a barrier is no indicator for CLDN23 being a sealing protein. Also epithelia containing charge-selective channel-forming claudins are barrier-forming for all other solutes.

Response: We have changed the indicated sentence to “Since CLDN23 was identified in the apical lateral plasma membrane corresponding to TJs in differentiated luminal IECs that express a number of anion and cation barrier-forming CLDNs, we hypothesized that CLDN23 could function as a barrier-forming CLDN.” Please refer to page 7 of the revised manuscript.

18. Throughout the text and Figures, CLDNs of numbers ## are written arbitrarily as "CLDN##" or as "CLDN-##", sometimes even within one sentence. In contrast to the full name "claudin-##", commonly no hyphens are used for "CLDN##".

Response: We have modified the manuscript text according to the reviewer’s suggestion (CLDN##).

19. Add hyphens, e.g. line 33: stem cell-derived, 174: CLDN23-depleted, 183: CLDN23-containing, 185 shRNA-mediated. Please check for all occasions throughout the text.

Response: We have incorporated the suggested edits.

Reviewer #3

The manuscript titled “Claudin-23 Regulates Epithelial Barrier Function by Altering Paracellular Pore Geometry” submitted for publication by Raya-Sandino et al describes the characterization of a novel claudin, CLDN23, whose function had been previously unknown. The manuscript is extremely well-written and the experiments are well designed, executed, and validated with proper controls and statistics. However, many claudins are spatially expressed and play a role in barrier function and although this was not known for CLDN23, the effect on paracellular pore size seems to be only an incremental increase in understanding the role claudins play in intestinal physiology. Others seem to have previously reported that claudins can effect pore size (Hempel et al 2022, Alberini et al 2017).

Response: We thank the reviewer for these positive comments and respectfully would like to point out the novelty of our study below.

Major comments:

1, 2. There was no prediction of a novel or unique role for CLDN23 and other hypotheses could have been explored such as a role in extracellular matrix interactions, proliferation, cell signaling, and differentiation. Of particular interest would have been characterization of the mechanisms that control the spatial expression of CLND23.

Response:

- We agree that recent studies (Alberini et al., 2017; Hempel et al., 2022) have highlighted the importance of charged pore-lining residues in determining geometry and ion charge selectivity. However, these previous studies focused on channels formed by a single claudin protein. How different CLDNs combine to alter channel

- architecture and the potential impact of such interactions on TJ permeability to ions and macromolecules as described in the current manuscript is novel.
- Here, we demonstrate for the first time that tight junction paracellular permeability can be dramatically altered by addition of a single claudin protein, in this case, claudin-23. This conclusion is based on complementary cell biologic, *in vivo* and computational modeling approaches. This work also represents the first rigorous examination of an uncharacterized and non-classic claudin protein in controlling intestinal epithelial barrier function.
 - We performed molecular dynamics simulations that revealed that heteromeric and heterotypic CLDN interactions alter pore architecture and overall net charge of pores with important functional consequences for ion permeability. Of note, we unexpectedly found that pore properties are not just regulated by claudin stoichiometry, since differently organized complexes with the same stoichiometry form pores with unique architecture.
 - Analysis of claudin heteromeric compatibility was validated using a novel computational model (PANEL) which calculates the free energy of all possible protein-protein interactions in a membrane bilayer. Computational modeling by PANEL has general applicability beyond claudins and can be used to characterize interactions between any pair of membrane proteins.
 - Our work supports an emerging concept that specific combinations of claudins are differentially expressed in mucosal epithelial cells during differentiation to spatially control barrier function. Our study utilizes physiologically relevant and sophisticated in-vivo, ex-vivo and in-vitro systems to demonstrate this concept.
 - We provide new insights in terms of how spatial expression of claudin family members in epithelial cells facilitates tighter barrier properties at the luminal surface of the gut compared with proliferative epithelial cells at the crypt base. Such differential claudin regulation plays an important role in regulating mucosal tissue homeostasis and is perturbed in inflammatory disease.
 - A detailed understanding of how combinations of claudin proteins controls flux across paracellular pores could facilitate rational design of therapeutics that can be delivered across epithelial barriers in the gut, lung and skin. Furthermore, knowledge gained from our study could facilitate design of therapeutic molecules that tighten the mucosal barrier in inflammatory states and improve disease outcomes.

We agree that in addition to regulation of barrier function, several intercellular junctional proteins control other processes including homeostasis and repair. Respectfully, we think that this is beyond the current scope of the current manuscript. Our future studies will investigate non-barrier forming functions of CLDN23.

3. Relating the loss of CLND23 expression to specific diseases would increase the relevance of the findings. There is no discussion of any phenotypic findings in the genetically modified mouse model and so are they assumed to be normal?

Response: We did not observe any change in baseline mucosal architecture in mice lacking *Cldn23* in colonic epithelial cells. We now include this data in the revised manuscript (**Extended data Fig. 5**, page 8). This manuscript describes the physiological role of CLDN23 in controlling

epithelial barrier function. Future studies will determine the role CLDN23 plays during inflammatory and neoplastic diseases.

4. *The discussion mainly reiterates the results without really discussing how the data significantly advances the field. What will be possible now that this data is known?*

Response: We have modified the discussion to better highlight how the current study advances the field as recommended by the reviewer.

5. *With the widespread use of human colonic organoids in the GI field, the use of cell lines in the paper seems outdated and not state of the art. Human colonic organoids can be genetically modified using CRISPR-CAS9 approaches and a series of experiments in this area would have greatly increased the impact of this paper.*

Response: Respectfully, for this manuscript we used cell lines as a complement to primary colonic epithelial cells obtained from genetically modified mice that were cultured as differentiated monolayers which is technically complex. We now show CLDN23 expression in primary human colonoids is similar to that observed in murine colonoids (**Fig. 1, E and F, page 6**). Therefore, we feel that *CLDN23* knockdown in human colonoids is beyond the scope of the current study.

6. *Some of the use of SEM seems inappropriate based on the samples size (less than 10).*

Response:

We have consulted with a biostatistician and, updated statistics throughout the revised manuscript are now presented as mean \pm standard deviation. We have also included individual data points in all graphs to better demonstrate data distribution.

References

- Alberini, G., Benfenati, F., & Maragliano, L. (2017). A refined model of claudin-15 tight junction paracellular architecture by molecular dynamics simulations. *PLoS One*, *12*(9), e0184190. <https://doi.org/10.1371/journal.pone.0184190>
- Berselli, A., Alberini, G., Benfenati, F., & Maragliano, L. (2022). Computational Assessment of Different Structural Models for Claudin-5 Complexes in Blood-Brain Barrier Tight Junctions. *ACS Chem Neurosci*, *13*(14), 2140-2153. <https://doi.org/10.1021/acchemneuro.2c00139>
- Daugherty, B. L., Ward, C., Smith, T., Ritzenthaler, J. D., & Koval, M. (2007). Regulation of heterotypic claudin compatibility. *J Biol Chem*, *282*(41), 30005-30013. <https://doi.org/10.1074/jbc.M703547200>
- Fuladi, S., McGuinness, S., & Khalili-Araghi, F. (2022). Role of TM3 in claudin-15 strand flexibility: A molecular dynamics study. *Front Mol Biosci*, *9*, 964877. <https://doi.org/10.3389/fmolb.2022.964877>

- Gonschior, H., Schmied, C., Van der Veen, R. E., Eichhorst, J., Himmerkus, N., Piontek, J., Gunzel, D., Bleich, M., Furuse, M., Haucke, V., & Lehmann, M. (2022). Nanoscale segregation of channel and barrier claudins enables paracellular ion flux. *Nat Commun*, *13*(1), 4985. <https://doi.org/10.1038/s41467-022-32533-4>
- Hempel, C., Rosenthal, R., Fromm, A., Krug, S. M., Fromm, M., Gunzel, D., & Piontek, J. (2022). Tight junction channels claudin-10b and claudin-15: Functional mapping of pore-lining residues. *Ann N Y Acad Sci*, *1515*(1), 129-142. <https://doi.org/10.1111/nyas.14794>
- Irudayanathan, F. J., Wang, X., Wang, N., Willsey, S. R., Seddon, I. A., & Nangia, S. (2018). Self-Assembly Simulations of Classic Claudins-Insights into the Pore Structure, Selectivity, and Higher Order Complexes. *J Phys Chem B*, *122*(30), 7463-7474. <https://doi.org/10.1021/acs.jpcc.8b03842>
- Krauss, D., Eisenberg, B., & Gillespie, D. (2011). Selectivity sequences in a model calcium channel: role of electrostatic field strength. *Eur Biophys J*, *40*(6), 775-782. <https://doi.org/10.1007/s00249-011-0691-6>
- Krieger, E., & Vriend, G. (2014). YASARA View - molecular graphics for all devices - from smartphones to workstations. *Bioinformatics*, *30*(20), 2981-2982. <https://doi.org/10.1093/bioinformatics/btu426>
- Land, H., & Humble, M. S. (2018). YASARA: A Tool to Obtain Structural Guidance in Biocatalytic Investigations. *Methods Mol Biol*, *1685*, 43-67. https://doi.org/10.1007/978-1-4939-7366-8_4
- Milatz, S., Krug, S. M., Rosenthal, R., Gunzel, D., Muller, D., Schulzke, J. D., Amasheh, S., & Fromm, M. (2010). Claudin-3 acts as a sealing component of the tight junction for ions of either charge and uncharged solutes. *Biochim Biophys Acta*, *1798*(11), 2048-2057. <https://doi.org/10.1016/j.bbame.2010.07.014>
- Nakamura, S., Irie, K., Tanaka, H., Nishikawa, K., Suzuki, H., Saitoh, Y., Tamura, A., Tsukita, S., & Fujiyoshi, Y. (2019). Morphologic determinant of tight junctions revealed by claudin-3 structures. *Nat Commun*, *10*(1), 816. <https://doi.org/10.1038/s41467-019-08760-7>
- Nalle, S. C., & Turner, J. R. (2015). Intestinal barrier loss as a critical pathogenic link between inflammatory bowel disease and graft-versus-host disease. *Mucosal Immunol*, *8*(4), 720-730. <https://doi.org/10.1038/mi.2015.40>
- Trott, O., & Olson, A. J. (2010). AutoDock Vina: improving the speed and accuracy of docking with a new scoring function, efficient optimization, and multithreading. *J Comput Chem*, *31*(2), 455-461. <https://doi.org/10.1002/jcc.21334>
- Van Itallie, C. M., & Anderson, J. M. (2011). Measuring size-dependent permeability of the tight junction using PEG profiling. *Methods Mol Biol*, *762*, 1-11. https://doi.org/10.1007/978-1-61779-185-7_1

REVIEWERS' COMMENTS

Reviewer #1 (Remarks to the Author):

The authors addressed my questions. I recommend this article for publication.

Reviewer #3 (Remarks to the Author):

The authors have significantly revised the manuscript titled “Claudin-23 Regulates Epithelial Barrier Function by Altering Paracellular Pore Geometry” and have addressed most of this reviewers’ major comments.

In the response to this reviewer’s comments, the authors provide a wonderful bulleted list of how their study is novel. This response is extremely well written and highlights the significant advancements and findings of the study in a concise and digestible manner. Although the discussion was much improved with several sections now highlighting their findings in the context of the field, this well written response should be modified and included as the first paragraph of the discussion instead of the lackluster recap that is currently in place. This modification will help the reader consolidate the findings in a very dense paper to the key messages that the authors would like the reader to take home. With the broad authorship of the journal, this will greatly add to reader satisfaction.

Reviewer 1

Comments to the authors: *The authors addressed my questions. I recommend this article for publication.*

Response: We thank the reviewer for recommending our article for publication.

Reviewer 3**Comments to the authors:**

The authors have significantly revised the manuscript titled “Claudin-23 Regulates Epithelial Barrier Function by Altering Paracellular Pore Geometry” and have addressed most of this reviewers’ major comments. In the response to this reviewer’s comments, the authors provide a wonderful bulleted list of how their study is novel. This response is extremely well written and highlights the significant advancements and findings of the study in a concise and digestible manner. Although the discussion was much improved with several sections now highlighting their findings in the context of the field, this well written response should be modified and included as the first paragraph of the discussion instead of the lackluster recap that is currently in place. This modification will help the reader consolidate the findings in a very dense paper to the key messages that the authors would like the reader to take home.

The broad authorship of the journal will greatly add to reader satisfaction.

Response: We thank the reviewer for this thoughtful comment. As recommended, we have modified the first paragraph of our discussion to reflect the significant advancements and findings of our study. Additionally, as recommended by the editor we have also modified our abstract and introduction to further increase the appeal of our paper to a broad readership.